# Exploring soccer transfers in Spanish League– The hidden role of strategic differences among teams

**Paulo Reis Mourao**[1], **Jesyca Salgado-Barandela**[2]*

**1** Deparment of Economics & NIPE, University of Minho, Braga, Portugal, **2** Deparment of Business Organization and Marketing, University of Santiago de Compostela, Santiago de Compostela, Galicia, España

* jesyca.salgado@usc.es

## Abstract

Transfers in the football world have become a hot topic in academic studies in recent years. Spanish league (La Liga) is one of the men's professional football leagues that have driven some of the most notorious transfers. In this way, we test determinants for the transfer relationships of football players in the Spanish League in the most expensive seasons with records (2018/2019 and 2019/2020 seasons). Furthermore, we identify determinants for the values of the observed transfers. The empirical analysis shows relevant findings. We recognize two implications. First, Spanish soccer transfers are not random and this evidence reflects the environment of imperfect competition characterizing La Liga. Second, the non-randomness of the transfer process can be associated with an increasing inequality among teams and with a threat to the competitive balance in professional sports.

## 1. Introduction

When a player moves from one club to another, his registration is transferred to that club, normally for a fee agreed between the two parties. This is the definition of football transfer, which involves two aspects: the transfer of the player from one team to another and the amount of money agreed between both teams. Transfers in the football world have become a hot topic in academic studies in recent years. The monetary resources involved as well as the growing trend helped in this highlight. The works focused on soccer transfers allow the identification of several lines of development. With no pretensions of exhaustion, we highlight three common lines of these publications. In a first line, there is the discussion of determinants for such flows, therefore determinants for transfers, either in number of players involved or in the associated amount [1–7]. In a second line, there are works like Mourao [2, 8] that seek to assess the efficiency of transfers considering their own restrictions on the part of the involved determinants. Finally, in a third line, the specificity of the transfers is evaluated in the consideration of certain realities, such as the places occupied by some athletes, the nature of the national championships involved or the relationship with certain observed periods [9–11].

**Funding:** The author(s) received no specific funding for this work.

**Competing interests:** The authors have declared that no competing interests exist

In a special way, the study of the determinants of transfers has achieved an important dimension in professional soccer. As Mourao [2] explains, the provisions of Financial fair play have pushed clubs to adopt transfer market strategies to save on relevant expenses and amortize transfers. At the same time, this strategy has increased the level of relevant income in terms of capital gains on player disposals and this effect is stronger for clubs from professional soccer leagues [12]. On the other hand, there are professional soccer leagues that show a notable increase in the monetary resources of player´s tranfers. This situation represents an opportunity to analyse the factors that influence transfers. Furthermore, scientific studies have evolved towards the consideration of new statistical tools in the study of determinants in this topic, such as network analysis [13].

One of the men's professional football leagues that have driven some of the most notorious transfers is the Spanish league (La Liga). In the last decade the average value per club in question has grown from € 15.23 million (from the season 2010/2011) to € 74 million in 2019/2020 [14]. If the nature of the Spanish league contributes to this value (we remember that it is traditionally considered one of the 3 most valued professional football leagues in world), other specific reasons contribute to this notoriety. Being a league that traditionally places clubs in the final stages of the main European club competitions creates a special attractiveness in its clubs for receiving the best players in the world [15]. Additionally, La Liga has become a league with millions of worldwide followers, with a growing number since 2000. Finally, the values of the brands associated with La Liga's clubs have reached record numbers over the last few seasons, with the Spanish championships having some of the teams with the highest value in the international market, regardless of the sport [16].

These reasons are strong reasons for an increasing interest in an analysis of the costliest seasons in terms of transfers. Thus, contributions can be made to the topic of study of the determinants of transfers in relation to the upward tendency of transfers in professional soccer leagues. In this way, the main objective of this work is testing determinants for the transfer relationships of football players in the Spanish League in the most expensive times with records. For it, we use network analysis and we test the hypothesis about dominant teams create hierarchical structures in which they move more players to lower-ranked teams rather than to other highly ranked teams. We use random graph models (ERGM) confronted with Panel-data Logit models to teste the determinants of the relationships. Thus, we will observe if certain teams' characteristics enhance the creation of ties with other teams in terms of players' transfers. Furthermore, we identify determinants for the values of the transfers observed and we use Ordinary Least Squares with Random Effects to estimate this model.

The main knowledge gap covered by this work is to study the dimensions responsible for the value of transfers from an approach of intense dynamics between clubs using network analysis. This work adds to the high-potential area of social network analysis in sports organizations [1, 9, 13, 17]. In this way, empirical analysis shows relevant findings. First, Spanish soccer transfers are not random and this evidence reflects the environment of imperfect competition characterizing La Liga. Second, the non-randomness of the transfer process can be associated with an increasing inequality among teams and with a threat to the competitive balance in professional sports. Additionally, we identify that larger point differences, as well as age of the team, balances of transfers and attendances throughout the seasons increase the value of the transfers observed.

The remaining structure of this paper is as follows. Section 2 reviews the literature in relation to attributes and determinants analysis to explain football transfers. Section 3 describes the data and explains the used methodology. Section 4 shows our empirical efforts regarding testing determinants of transfer networks and identify determinants for the values of the transfers observed. Section 5 concludes and provides opportunities for further research.

## 2. Literature review: Determinants of transfers and the importance of implementing network analysis

Previous literature has investigated the main drivers of transfer fees paid by clubs. The evidences indicate that, the level of a transfer fee is linked to the player's characteristics and performance features [6, 7, 14, 15]. Franceschi et al. [16] identifies in his systematic review study that age and age squared are the most often tested variables, consistently associated with significant. In addition to age, variables such as usage of the player (minutes played or number of appearances) and international status and decisiveness (goals and assists) are frequently included in the studies and positively and consistently influence the valuation of the football player. On the other hand, Franceschi et al. [16] explains that this positive relationship can be explained by the low-score nature of football. When the usage variables are studied together with the performance variables, the significant results are low. Furthermore, these indicators are probably correlated, thus they may weaken each other's influence on the dependent variable.

Previous studies also consider the economic-financial dimension in the analyzed determinants. In a specific way, economic variables are defined such as wage rate, relevant incentives, and the remaining duration of the contract, play a role in determining the transfer fee for player registration [5, 17]. In relation to the financial dimension, variables such as the values of past transfers and the respective balances, in addition to the financial indicators observed from the Account Reports of the Clubs [2, 7]. Campa [5] explains that they reflect the influence of factors related to neither players' skills nor to other economic features, for example: the race of a player or the number of Google searches associated with the player.

In relation to the dependent variable, most of the works in extant literature propose at least one model specification where the dependent variable is the transfer fee. It is necessary to consider that transfer fee it is not available for every player at any time but rather only when a transfer eventually occurs [12, 16]. For this reason, studies of transfer determinants usually use crowdsourced estimates of players value, retrieved from the website Transfermarkt [16]. It is possible to find differences in relation to the use of transfer fee as a dependent variable. Matesanz et al. [18] use three variables to construct transfer data: transfer spending (transfer spending is the amount of money each club has spent in transfers in each season), transfer earnings (represent the amount of money each club has generated from transfers in each season) and transfer fee volume (transfer spending + transfer earnings). On the other hand, Liu et al. [19] and Rosseti and Caproni [20] detail transfers by type of transaction: free / payed transfer, and loans. In the case of Coates et al. [21] they treat loans also as a connection between clubs.

In the literature, only a small number of papers have analyzed football transfers using network analysis. Liu et al. [19] studied the relationship between the success of a club and player transfers on a dataset of transfer records from 2011 to 2015 of 410 professional clubs in 24 world-wide top class leagues. They show "that clubs' match performance and profitability from the transfer market are strongly associated with the coreness and brokerage properties of their corresponding nodes in the player transfer network" [19]. Moreover, Matesanz et al. [18] explore the evolution of the football players' transfer network among 21 European first leagues between the seasons 1996/1997 and 2015/2016. They get findings similar to Liu et al. [19]. Rosseti and Caproni [20] also studied the impact of club transfer strategies on team performance, but with a different database. In this case, they worked with data covering 25 years of trading among clubs of the 6 FIFA federations. Unlike Liu et al. [19], Rosseti and Caproni [20] found support for a nonlinear relationship between network characteristics and roster changes.

On the other hand, Coates et al. [21] extend the work previously done using a massive dataset on transfers and club characteristics to study the relation between transfer strategy and

performance. They use data on 23220 clubs from 189 countries from 1996 through 2016 and they found that a wide transfer network is harmful for a club's sport performance, and the best strategy is to have the lowest number of connections inside the league. As Coates et al. [21] explains "for financial performance, it is better to have a high number of international connections with clubs that have widespread transfer networks, whereas connections inside domestic league are not important".

More recently, Clemente y Cornaro [13] have applied network analysis to study number of transfers and the related costs for football players in the world market. They use all transfers between football clubs regarding the season 2020–2021, taking into account both winter and summer sessions, 44,481 transfers in total. They find that clubs of top football leagues tend to behave in a similar way in the network and to trade players each other and highlighted the role of several European countries that represent top leagues in the soccer market and are also involved as central countries in the economic trades.

In relation to those studies [13, 18–21] it is identified that the number of points is a variable commonly used as an independent or control variable. Liu et al. [19] use Kendall's Tau coefficient to analyze the relationship between club functionalities versus network properties in international and domestic transfer networks, node properties in transfer network and loan network, and node properties in money leagues and farm leagues. These authors consider two variables in relation to the number of points, which are: average league points (domestic league) and IFFHS Club World Ranking points (The IFFHS Club World Ranking points are composed by the International Federation of Football History & Statistics (IFFHS) based on a set of rules composing both domestic and international match results). While, Matesanz et al. [18] include domestic rank and UEFA points to study the football player transfer market activities among European first leagues from 21 countries. Rosseti and Caproni [20] and Coates et al. [21] only consider points per game like a dimension to represent number of points.

In this same line, previous papers also consider characteristics of the players in the selection of independent or control variables. Matesanz et al. [18] and Rosseti and Caproni [20] include various characteristics in relation to the transferred players. In the case of Matesanz et al. [18] variables are: age, nationality, field position. Matesanz et al. [18] find that "the connection between transfer spending and sportive performance, especially in UEFA competitions, is extremely strong particularly for clubs from the top cluster, which might further limit the overall level of uncertainty of outcome". While, Rosseti and Caproni [20] include role, age and market value to players transferred, and they identify among other findings that roster stability plays a crucial role on the obtained results and more you change more unpredictable your future results will be. Coates et al. [21] include player´s rating in its methodological model. Also, it is the only work that also measures the performance of the coach using fixed effects for its measurement. Finally, the attendance variable is only included by Coates et al. [21], the rest of previous works do not consider it in its measurement.

## 3. Data and methodology

### 3.1 Data

The seasons studied here– 2018/2019 and 2019/2020 –were the most notorious Spanish seasons in terms of transfer values. The data proves that there have never been such high values in terms of transfers paid by Spanish club; also this period has allowed Spanish clubs the highest values received from transfer moves. In this way, we collected a database with transfers of the Spanish League (La Liga) for 63 weeks, between 1/July/2018 and 1/September/2019. The information to build the database has been obtained from the website transfermarkt.es. The descriptive analysis of the database shows that in the 63 weeks between 1/July/2018 and 1/

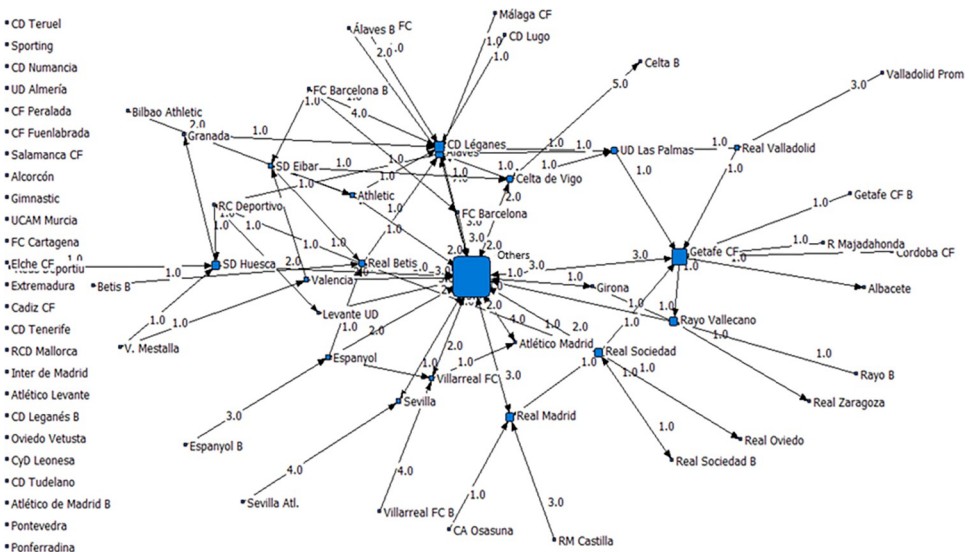

**Fig 1. Spanish transfer network—All transfers included (1ˢᵗ September 2019).**

September/2019 a total of 503 transfers had been reported, with a total value of € 3,471.94 million. In the case of the 2018/2019 season, 265 transfers occurred, with a total value of €1,630.67 million. On the other hand, during the first two months of the 2019/2020 season (July and August), 238 transfers occurred, with a total value of €1,841.27 million.

Just for illustrative purposes, we constructed Fig 1, which highlights these flows. Let us remark that the node "Otros" include all the other teams (of all championships in the globe) having had transfers with Spanish teams.

Thus, three reasons emerge for us to focus our study on these times, without neglecting to extend to more remote times. These three reasons are: the maximum achieved in relation to sports transfers in Spain, the emerging role of network analysis to study competitive phenomena, and the importance of observing the dimensions of oligopolies that exist in sport. First, the maximums observed during the transfer windows listed here, due to the current pandemic, have not yet been exceeded in the most recent transfer windows. It is important to understand what happens in these seasons because this provides relevant findings to explain a scenario of upward tendency in transfers. This situation is not exclusive to the Spanish Soccer League; other leagues also have high transfer costs. Thus, the understanding of a general situation of the transfer market can be improved.

Second, the transfer market has not considered so far either the role of the networks involved or the importance of more likely transfers due to imbalances in characteristics between clubs. Thirdly, behaviors induced in oligopolistic environments, such as that observed in professional soccer championships, induce asset transfers between companies that compete for different market shares, a central hypothesis in our study.

Our dependent variable for the model proposed in this study is the value of transfers from club A to club B at the observed seasons. The independent variables are the variables shown in Table 1, which follow the quoted literature from Section 2. In addition to the variables in Table 1, certain attributes are added to the model to explain the properties of the network. These attributes are: number of points obtained at the end of season, transfer balance value (millions of euros), average age of the team squad for the seasons and total attendance at the team's stadium for the seasons.

**Table 1. Descriptive statistics of the La Liga teams' attributes (2017–2018 and 2018–2019).**

| | Mean | Std Dev | Max | Min |
|---|---|---|---|---|
| number of points got at the end of season 2015/2016 | 55,12 | 17,489 | 91 | 32 |
| number of points got at the end of season 2016/2017 | 55,56 | 19,894 | 93 | 20 |
| number of points got at the end of season 2017/2018 | 55,72 | 17,516 | 93 | 20 |
| average number of points for the seasons 2015/2016, 2016/2017, and 2017/2018 | 55,47 | 14,068 | 91,3 | 35 |
| TrfBalan value (millions of Euros) at the end of season 2015/2016 | -1,115 | 26,861 | 78,9 | -90,95 |
| TrfBalan value (millions of Euros) at the end of season 2016/2017 | -0,453 | 37,797 | 88 | -142 |
| TrfBalan value (millions of Euros) at the end of season 2017/2018 | -4,135 | 27,132 | 47,1 | -95,6 |
| average TrfBalan value (millions of Euros) for the seasons 2015/2016, 2016/2017, and 2017/2018 | -1,901 | 20,636 | 26,7 | -76 |
| Average Age of the team squad for the seasons 2015/2016, 2016/2017, and 2017/2018 | 26,023 | 0,923 | 28,11 | 24,54 |
| Total Attendance at team's stadium for the seasons 2015/2016, 2016/2017, and 2017/2018 | 466.313 | 337.168,495 | 1.342.982 | 100.544 |
| Average Attendance per match at team's stadium for the seasons 2015/2016, 2016/2017, and 2017/2018 | 24.458 | 18.129,9161 | 73.159 | 5.292 |

Source: Author´s elaboration

The explanatory variables of our model were inspired by the literature presented in the previous section. Still, we recall the underlying reason as well as the supporting literature below.

The number of points in previous seasons is a variable associated with the value of the team's budget as well as the ability to generate transfers around the club. The use of values from several epochs allows for a more robust associated reading [18, 19]. The average value of the number of points allows you to control possible fluctuations outside the predictable performance of each team.

The variable focused on the transfer balance is a variable that shows the difference between the amount received from transfers made and the amount paid due to transfers that entered the club. In line with the literature consulted [18], higher transfer balances produce a leverage effect on the club's future operations. Once again, the presence of a variable associated with the average value (in this case, the balance of transfers) aims to provide greater robustness to the model. The average age of the team shows the need to renew the squad in future exercises. As suggested by Samur [22], aging teams increase the probability of incoming transfers in the future.

The total number of spectators reported for each team's home games is a variable that controls the number of points as well as the balance of transfers. Teams with good home game attendances show significant support from the surrounding community. The inclusion of the average attendance at home aims to control possible imbalances that consecutive fluctuations in attendance may suffer [23].

## 3.2 Methodology

Recalling the literature on network analysis, specifically Mariotti and Haider [24] and Pardalos and Zamalaev [25], we want to observe the selectivity of these transfers, testing the claim that certain teams make the moving of their players to other teams easier. Therefore, we want to observe if certain teams' characteristics enhance the creation of ties with other teams in terms of players' moves (or if these characteristics make these ties more difficult). We are particularly concerned with team budgets, points/ranks usual in championship or centrality positions [26, 27]. Therefore, the empirical development is divided into three stages. In the first place (section 3.2.1), we want to test the hypothesis about "Dominant teams" creating hierarchical structures in which they move more players to lower-ranked teams rather than to other highly ranked teams [28]. Our empirical section will test this hypothesis following the works of

**Table 2. ERGM estimates for La Liga players' transfers (2018–2019 and 2019–2020).**

| Parameters | Posterior Mean Estimate -2018 (Std Deviation) [Acceptance Rate] | Posterior Mean Estimate -2019 (Std Deviation) [Acceptance Rate] |
| --- | --- | --- |
| Model 1: Arc Baseline | | |
| Arc | -5.291* (0.247) [0.50] | -3.402* (0.095) [0.44] |
| Model 2: Structural Parameters | | |
| Arc | -5.891* (0.277) [0.52] | -3.245* (0.102) [0.42] |
| Reciprocity | 0.666* (0.288) [0.52] | 0.684* (0.249) [0.42] |
| Model 3: Attributes Model | | |
| Arc | -5.804* (0.139) [0.40] | -3.632* (0.127) [0.39] |
| Points16_DifferenceA | -3.310* (0.181) [0.40] | -4.118* (0.174) [0.39] |
| Points17_DifferenceA | -6.581* (0.407) [0.40] | -5.487* (0.218) [0.39] |
| Points18_DifferenceA | -0.438 (0.469) [0.40] | -0.088 (0.188) [0.39] |
| TrfBalan16_DifferenceA | 37.54* (0.160) [0.40] | 37.70* (0.199) [0.39] |
| TrfBalan17_DifferenceA | 27.22* (0.421) [0.40] | 28.12* (0.287) [0.39] |
| TrfBalan18_DifferenceA | 23.91* (0.248) [0.40] | 23.17* (0.472) [0.39] |
| Team_age_DifferenceA | -0.574* (0.213) [0.40] | -2.627* (0.222) [0.39] |
| Assistance_DifferenceA | -0.117* (0.007) [0.40] | -0.133*(0.052) [0.39] |
| Model 4: Combined Model | | |
| Arc | -5.704* (0.139) [0.50] | -2.911* (0.271) [0.40] |
| Reciprocity | 0.671* (0.362) [0.50] | 0.644* (0.352) [0.40] |
| Points16_DifferenceA | -3.300* (0.181) [0.50] | -4.108* (0.174) [0.40] |
| Points17_DifferenceA | -6.582* (0.407) [0.50] | -5.488* (0.218) [0.40] |
| Points18_DifferenceA | -0.338 (0.469) [0.50] | -0.088 (0.188) [0.40] |
| TrfBalan16_DifferenceA | 37.44* (0.160) [0.50] | 37.70* (0.199) [0.40] |
| TrfBalan17_DifferenceA | 27.21* (0.421) [0.50] | 28.09* (0.287) [0.40] |
| TrfBalan18_DifferenceA | 22.91* (0.249) [0.50] | 22.47* (0.411) [0.40] |
| Team_age_DifferenceA | -0.584* (0.214) [0.50] | -2.697* (0.232) [0.40] |
| Assistance_DifferenceA | -0.217* (0.107) [0.50] | -0.133*(0.051) [0.40] |

* means convergence statistics lower than 0.1. Note: for each attribute, we have also estimated the following dimensions: Value of Sender; Value of Receiver; Sum of both values; Product of both values. As the related estimates revealed themselves to be non-significant, we omitted them (although they are available upon request).

Hanneman and Riddle [29], Holland and Leinhardt [30], and Shumate and Palazzolo [31]. We start the empirical analysis using network analysis. The main empirical outcomes from this first stage will be summarized in Table 2.

Afterwards, we will also want to test the robustness of these outcomes consider panel data logit models (check the theoretical motivation at sub-section 3.2.2 and the empirical outcomes at sub-section 4.2). Finally, taking advantage of the panel data, we will test determinants for the different values of the transfers by recurring to Panel-data linear models (check Table 4 at the empirical sub-section 4.3 and the theoretical motivation at Sub-section 3.2.3).

**3.2.1 Exponential random graph models (ERGM).** As Morris [32] states, "[actors/teams] may come and go, the links form and dissolve, but the structural pattern remains." Thus, it is relevant to study the probability of observing a graph (i.e., a set of relationships) y on a fixed set of nodes. In our case, we keep the motivation to explain the existence of the observed structure of players' moves because of dimensions such as the existence of arcs, the simultaneous presence of bidirected moves (i.e., players coming from a team and going to another team and the reverse), or even the effects of certain exogenous variables, such as the points of each team or its budget value.

Following Holland and Leinhardt [30] and their original models, we will use exponential random graph models (ERGM). As these authors demonstrated, there are properties of ERGM that must be noted, as these types of models "can easily accommodate other relations, attributes, and structural estimates as predictors of a given network" [33]. Additionally, and still quoting Snijders et al. [33], ERGM can differentiate "the senders of relations, the receiver of relations, or that actors with the greatest differences in valued attributes are likely to have relations."

Following the notation of Shumate and Palazzolo [31] and Mourao [34], we can enunciate the ERGM model as follows:

$$P(X = x) = \frac{\exp\{\theta' z(x)\}}{k(\theta)} \tag{Eq 1.1}$$

According to Eqs 1.1, 1.2, we are modeling the probability of a given network P(x) depending on the estimation of a vector of model parameters ϴ. z(x) refers to a vector of network statistics, and k is a normalizing function. The role of this normalizing function is to guarantee a certain probability distribution across the random networks.

For the estimation of Eqs 1.1, 1.2, we have to use maximum pseudo likelihood estimations. This method fits a logistic regression for the vector of network statistics. Therefore, we will work with the following maximum pseudo likelihood estimation:

$$PL(\theta) = \prod_{i \neq j} \prod_{m=1}^{r} P(X_{ijm} = 1 | X_{ijm}^C)^{X_{ijm}} P(X_{ijm} = 0 | X_{ijm}^C)^{1 - X_{ijm}} \tag{Eq 1.2}$$

Following Shumate and Palazzolo [31], the maximum pseudo likelihood estimation for each parameter is computed as a product of the log-odds ratio of the following two probabilities: the probability for each observed tie, P(X = 1) and the probability of not observing that tie in the network, P(X = 0). Additional features–such as the use of pseudo likelihood ratio statistics, the discussion of the independence assumptions, and robustness in small samples–are discussed by authors including Shumate and Palazzolo [31] and Robins, Pattison and Wasserman [35]. Given the nature of our sample [36], we opted for estimating Eqs 1.1, 1.2 by Bayesian procedures. We follow Caimo and Friel [36] who state that "Instead of obtaining the point estimates [. . .], the Bayesian estimation generates the posterior distributions of the model parameters. In lieu of MLEs and standard errors, point estimates and measures of uncertainty are calculated as averages and standards errors of this distribution, respectively".

**3.2.2 Panel data logit model for robustness.** In order to explore the robustness of the results obtained by ERGM, we intend to estimate the following "Panel-data logit Model" (Eq 2.1) which will explain the probability of a transfer happening between each pair of La Liga's teams. Generically, this model has the following specification, for n periods and i observations [37]:

$$\Pr(Y_{i1}, \ldots, Y_{ini} | X_{i1}, \ldots, X_{ini}) = \int_{-\infty}^{\infty} \frac{e^{-v_i^2/2\sigma_v^2}}{\sqrt{2\pi}\sigma v} \left\{ \prod_{t=1}^{ni} F(y_{it}, X_{it}\beta + V_i) \right\} dv_i \tag{Eq 2.1}$$

In Eq 2.1, we model the probability of a given pair of teams I at a given season n having had a transfer of players. The random effect $v_i$ follows a Normal Distribution N (0, Sigma^2_v).

The following specification is also observed.

$$f(y, z) = \begin{cases} \dfrac{1}{1 + \exp(-z)} & \textit{if } y \neq 0 \\[2ex] \dfrac{1}{1 + \exp(z)} & \textit{otherwise} \end{cases} \qquad \text{(Eq 2.2)}$$

Finally, the panel-data likelihood l_i is specified as

$$l_i = \int_{-\infty}^{\infty} \frac{e^{-v_i^2/2\sigma_v^2}}{\sqrt{2\pi}\sigma v} \left\{ \prod_{t=1}^{ni} F(y_{it}, X_{it}\beta + V_i) \right\} dv_i \equiv \int_{-\infty}^{\infty} g(y_{it}, x_{it}, v_{it}) dv_i \qquad \text{(Eq 2.3)}$$

**3.2.3 Panel data linear models for transfers´values.** After these steps, we will also test determinants for the values of transfers observed (at subsection 4.3). Our dependent variable is thus, for this model, the value of transfers from club A to club B at the observed seasons. To estimate the model relating to the value of transfers made between club A and a different destination, and as we are dealing with panel data, we used two estimation methods: Ordinary Least Squares with Fixed Effects and Ordinary Least Squares with Random Effects. To choose between the two methods, we performed the Hausman test. The result of the Hausman test always had a p-value greater than 10%. This value suggests that random effects should be used, revealing that each club and each season have introduced significant effects of their own. More detailed results will be displayed upon request.

## 4. Empirical analysis

### 4.1 Hypotheses for determinants of transfer networks

When we intend to analyze network structures, studies are based on the need to explore not only the dimensions that explain the values observed in flows but also on the need to explore dimensions that explain the existence and meaning of flows in the network.

Thus, we have two groups of hypotheses for the design observed in a network:

- structure hypotheses, where arcs/connection pairs are evaluated, among other more complex structures;

- and the hypotheses of the determinants of values, closer to linear estimation analysis.

Within the second group, we cite the literature that had already been discussed before—[2, 28, 29, 34]. Therefore, for player transfers in the Spanish League, we will analyze sporting performance (measured by the team's number of points in previous seasons) but also the financial performance of that period (evaluated by two variables–the team's budget and the balance of transfers). But we will also evaluate, for each pair of teams, the respective differential of these variables, as is common in the literature on network analysis. Within the first group, the structure hypotheses, network analysis suggests the assessment of the existence of arcs as well as reciprocal flows [31, 32].

First, we show the result obtained from the ERGM model. Following Shumate and Palazzolo [21], if the estimated posterior mean parameters have significant values, i.e., higher than twice the estimated standard deviation, then this is an indication of a good fit. Additionally, the acceptance rate must be as high as possible. In empirical terms, posterior distributions are a 'compromise' between two sources of information–the likelihood estimation and the prior distributions of the parameters–which has been found empirically adequate for studying social networks [25]. We included the most-frequently cited indices for goodness-of-fit of ERGM

estimated by Bayesian procedures in Table 2. Following Caimo and Friel [26], the options regarding Bayesian estimation must be assessed by evaluating the acceptance rates of the posterior distribution estimates refer, "acceptance rate as high as possible" and by checking the statistical significance of the posterior mean estimates [26]. Table 2 exhibits the estimates and indices for the goodness-of-fit for each model.

In Table 2, we started with only one parameter, i.e., with the arc parameter. The estimate (of negative sign) suggests that the existence of ties in the observed network is less likely to be explained by chance. Therefore, our idea regarding the intentionality/influence of explicative dimensions on the transfers in Spanish La Liga is reinforced.

Nevertheless, in Table 2, we explored a second model in which we included structural parameters. In addition to arcs, we added "reciprocity". "Reciprocity", in network analysis, means that we consider mutual relations (in our network, two teams simultaneously receiving and sending players from one to another).

Then, we have a third model. This model includes the possibility of certain attributes explaining the properties of the network. Following the literature [2, 11, 38, 39], we observed the following attributes for each team of the tie:

- Number of points obtained at the end of season 2015/2016;

- Number of points obtained at the end of season 2016/2017;

- Number of points obtained at the end of season 2017/2018;

- Transfer balance value (millions of euros) at the end of season 2015/2016;

- Transfer balance value (millions of euros) at the end of season 2016/2017;

- Transfer balance value (millions of euros) at the end of season 2017/2018;

- Average age of the team squad for the seasons 2015/2016, 2016/2017, and 2017/2018;

- Total attendance at the team's stadium for the seasons 2015/2016, 2016/2017, and 2017/2018.

The reasons for this choice follow the established sports economics literature. A team's competitiveness can be measured by the track of its number of points along the seasons, as well as the profile of its rank. So we collected dimensions such as the points at each season's end [2, 40]. The financial aspect is of crucial relevance in this analysis, so we collected the values for the transfer balance identified for each team during the observed seasons (difference between the revenues received from the sales of players and the expenses from new players' acquisitions). The sources for these variables are our own computations of official values for each team's points, [39]. For the transfer balance, we referred to Barajas and Rodriguez [38] and Mourao [2].

The attributes in social analysis can be extended to four usual variables–Sender (of the Transfer in our case, the 'Seller' team), Receptor (of the Transfer, the 'Buying' team), Sum of both values, and the Difference between the values of the Sender to those of the Receptor. We analyzed all of these variables, which resulted in a list of 44 variables (11 attributes multiplied by the four usual dimensions). However, in Table 2, we only keep the statistically significant ones, which are all those related to the differences between the sending team and the receiving one. The full estimates are available upon request. Therefore, we can state that more important than absolute values for explaining the soccer transfers–as previously hypothesized by Mourao [2]–we posit that transfers tend to occur between differently ranked teams, which converges to a validation of our Hypothesis–that dominant teams create hierarchical structures in which

they move players to lower-ranked teams easier than to other same-ranked teams to keep themselves (the Dominant teams) competitive.

Following Musau [41], the match of values is used to test the homophily among the network's participants, and the mismatch of the values is used to test the likelihood of ties among different individuals. As Barajas and Rodriguez [38] or Mourao [2] observed, soccer transfers' relationships (e.g., among teams) tend to be characteristic of oligopolist markets. Therefore, authors such as Chaudhuri, Benchekroun and Breton [42] claim that the financial flows among the oligopolists themselves are less frequent (or characterized by a low significant value) than between an oligopolist and other (following) firms or companies. In the professional soccer world, this means that we can expect a reduced number of transfers between top-ranked teams (such as a player being transferred from FC Barcelona to Real Madrid) than between a highly competitive team and not-so-competitive teams. Therefore, in the ERGM attribute model, we have to especially consider the mismatch of the considered attributes.

As we have noted, and following Shumate and Palazzolo [31], in addition to the relevance of obtaining significant estimates for the parameters, also related to the relationship between the estimated value and estimated standard error (which must be greater than 2.0 for significant estimates), we must also observe the values obtained for the goodness-of-fit indices (namely, the Acceptance Rates). Following the literature (detailed by Robins et al. [35]) and regarding the goodness-of-fit indices for the models, it is important to note some details. For instance, in Table 2, for 2018, we estimated the arc parameter (ML estimate: -5.291* (0.247) [0.50]; standard error 0.247; Acceptance rate 0.50). Considering these network parameters of Model 1 in Table 2, we can conclude that the negative estimated values for the arc parameters in Table 2 shall be interpreted following Frank and Strauss [43]; there is an estimated negative endogenous propensity for the parameter existing in the network. In other words, this indicates that ties are less likely to be observed in the Spanish transfers' networks than by chance. This suggested an additional stimulus to search for exogenous dimensions to explain the existence of the observed ties.

Regarding Model 2, we included one structural parameter in addition to arcs, i.e., reciprocity. The maximum likelihood (ML) estimates for these latter parameters have positive signs, meaning the presence of a positive propensity for this parameter in the network. This means that two random Spanish teams tend to 'exchange' players at the end of each transfer window (not only selling/buying players to the other node). The exhibited goodness-of-fit indicator (the Acceptance Rate) shows a significant improvement if compared to Model 1.

In Model 3, we have an attribute model in which we added network attributes to the opportunity to explain the network's properties. These (attribute) parameters are considered independent of the structural parameters [31]. In addition to the confirmed convergence of the ML estimates, the goodness-of-fit indices are also highly satisfactory for the estimated parameters of arc and reciprocity. Nonetheless, we can suggest that some attributes–such as team points' or transfer balance's mismatch–tend to create ties among teams regarding players' moves. In other words, it is more likely that a player moves from one team to another if these teams were distant in terms of points at the end of the previous seasons, suggesting that teams having the same competitive output tend to avoid reciprocal moves of their players.

Finally, in Model 4, we have a combined model with structural parameters and attributes. This model provides the convergence statistics that are good for the ML estimates and satisfactory values for the goodness-of-fit of this model's indices, even for the estimated structural parameters.

Therefore, we can provide a more complete perspective with Model 4. In this model, we observe that the ties between teams are not random (as suggested by the estimated negative sign for the ML estimate related to arc). We also find that there are more ties between teams

with reciprocal moves than with unilateral moves of players. We can further assert that players tend to move between teams that are not of the same competitive rank (e.g., it is more likely that a player moves from a bottom-ranked team to a top-ranked team than moving to another bottom-ranked team). Independent of the used variable, this direction remains. Conversely, moves are less likely between teams of the same championship or same budget, also reinforcing the competitive environment characterizing Spanish transfers.

Comparing the estimates for 2018 with those for 2019 in Table 2, we do not observe significant distinctions. This evidence highlights the relevance of the estimated parameters for describing the Spanish networks of transfers, independent of the season and the direction of the found significant attributes–namely, the difference between points or between transfer balances for increasing the likelihood of a transfer tie.

In summary, we can argue that Spanish soccer transfers are not random and they can be explained essentially by transfers established between teams of different ranks of sporting competitiveness. This means that transfers between two Spanish teams that have a significant difference in terms of points at the end of the season are reported more frequently than between two teams with close standings. Additionally, teams tend to establish transfers with other teams with squads whose average age is similar to their own average age and with close assistances. This evidence reflects the environment of imperfect competition characterizing La Liga, which leads to this kind of strategical behavior in the transfer markets.

## 4.2 Testing determinants of transfer networks through panel logit models

In order to explore the robustness of the previous results, we intend to estimate the following "Panel-data logit Model". Table 3 shows the results obtained, considering different specifications of the simultaneous presence of the determinants tested for the ERGM. Remember that, as a dependent variable, we will have a binary variable where the value of 1 identifies the transference from club A to club B and 0 if there was no transfer, at that time, from club A to club B. It was considered as a different observation the transfer of club B to club A at the same season (also marked by 1 in positive case or 0 in null case). Therefore, we had 882 observations built for this robustness step.

In the first specification, in Table 3, we include only the difference in points over the previous seasons between the teams involved in the transfers, the points of the player's acquiring team in the respective season, the difference in transfer balances between the teams involved, and the balance of transfers from the acquiring team. Given the structural parameter of Arc being proper of networks, we will not include Arc in the specifications for the panel data. For reasons of multicollinearity, neither the transferring team's points nor the transferring team's transfer balance are included in the specification. In the second specification, in addition to the previous variables, we also included the age differences between clubs and the club age of the acquiring team [2]. Finally, in the third specification, we added to the previous specification the average assists level of the acquiring team as well as the difference between the average assists level between the teams involved in the transfers.

As usual to assess the quality of the estimation in panel data with binary dependent variables, we included, for all specifications, the value of Log-likelihood, Correctly classified observations (%), and Pseudo-R2, as well as the value of Rho (the panel-level variance component).

In general, the results presented in Table 3 converge with the results proposed by the estimation of the ERGM model (Table 2). It should be noted that additionally Zero-Inflated Poisson and ZINB (Zero Inflated Negative Binomial) models were estimated with convergent results (details will be displayed if requested).

**Table 3. Panel logit random effects estimates for La Liga players' transfers (2018–2019 and 2019–2020).**

|  | Logit estimates | Marginal effects |
|---|---|---|
| Points16_Difference | -0.184** (0.084) | -0.0634 (0.033) |
| Points16_Buying team | 0.426 (0.362) | 4.102 (3.171) |
| Points17_Difference | -0.417* (0.181) | -0.587* (0.217) |
| Points17_Buying team | 0.545* (0.247) | 0.088 (0.058) |
| Points18_Difference | -0.336 (0.269) | -0.266** (0.076) |
| Points18_Buying team | -0.227* (0.157) | -0.377* (0.157) |
| TrfBalan16_Difference | 0.378*** (0.068) | 0.289*** (0.088) |
| Trfbalanc16_Buying team | 0.299** (0.121) | 0.229* (0.149) |
| TrfBalan17_Difference | 0.221* (0.121) | 0.171*** (0.041) |
| TrfBalan17_Buying team | 0.102*** (0.022) | 0.063* (0.033) |
| TrfBalan18_Difference | 0.245* (0.114) | 0.134** (0.054) |
| TrfBalan18_Buying team | 0.115* (0.071) | 0.135** (0.055) |
| Seasons dummies | Yes |  |
| N | 882 |  |
| Log-likelihood | 153.857 |  |
| Correctly classified (%) | 0.393 |  |
| Pseudo-R2 | 0.320 |  |
| Points16_Difference | -0.181** (0.085) | -0.061 (0.035) |
| Points16_Buying team | 0.256 (0.269) | 2.122 (3.172) |
| Points17_Difference | -0.432** (0.131) | -0.583* (0.213) |
| Points17_Buying team | 0.544* (0.244) | 0.085* (0.015) |
| Points18_Difference | -0.336 (0.269) | -0.266** (0.076) |
| Points18_Buying team | -0.273* (0.157) | -0.370* (0.157) |
| TrfBalan16_Difference | 0.388*** (0.068) | 0.289*** (0.089) |
| Trfbalanc16_Buying team | 0.280** (0.101) | 0.207* (0.140) |
| TrfBalan17_Difference | 0.221* (0.121) | 0.171*** (0.041) |
| TrfBalan17_Buying team | 0.125*** (0.024) | 0.367* (0.033) |
| TrfBalan18_Difference | 0.274 (0.414) | 0.143** (0.054) |
| TrfBalan18_Buying team | 0.115 (0.075) | 0.135** (0.055) |
| Team_age_Difference | 0.112** (0.052) | 0.118 (0.082) |
| Team_age_Buying team | 0.193* (0.093) | 0.199 (0.099) |
| Seasons dummies | Yes |  |
| N | 882 |  |
| Log-likelihood | 318.327 |  |
| Correctly classified (%) | 0.461 |  |
| Pseudo-R2 | 0.362 |  |
| Points16_Difference | -0.185** (0.085) | -0.0644 (0.035) |
| Points16_Buying team | 0.456 (0.369) | 4.108 (3.174) |
| Points17_Difference | -0.412* (0.181) | -0.588* (0.218) |
| Points17_Buying team | 0.544* (0.247) | 0.088* (0.018) |
| Points18_Difference | -0.330 (0.259) | -0.226** (0.075) |
| Points18_Buying team | -0.223* (0.153) | -0.370* (0.159) |
| TrfBalan16_Difference | 0.374*** (0.069) | 0.289*** (0.087) |
| Trfbalanc16_Buying team | 0.289** (0.121) | 0.227* (0.141) |
| TrfBalan17_Difference | 0.225* (0.129) | 0.172*** (0.042) |
| TrfBalan17_Buying team | 0.105*** (0.004) | 0.067* (0.032) |
| TrfBalan18_Difference | 0.275* (0.114) | 0.133** (0.051) |

(*Continued*)

**Table 3.** (Continued)

| | Logit estimates | Marginal effects |
|---|---|---|
| TrfBalan18_Buying team | 0.117* (0.071) | 0.135** (0.053) |
| Team_age_Difference | 0.115** (0.059) | 0.144** (0.061) |
| Team_age_Buying team | 0.199* (0.092) | 0.098* (0.048) |
| Assistance_Difference | -0.287**(0.147) | -0.166* (0.072) |
| Assistance_Buying team | 0.377 (0.301) | 0.254 (0.0182) |
| Seasons dummies | Yes | |
| N | 882 | |
| Log-likelihood | 353.877 | |
| Correctly classified (%) | 0.483 | |
| Pseudo-R2 | 0.380 | |

Robust standard errors between parentheses. Significance level: 1%, ***; 5%, **; 10%, *.

Thus, marked differences in points observed in previous seasons reduce the probability of having a transfer considering the clubs observed. However, clubs with more points tend to be associated with higher odds. The balances of transfers from the past also influence the probability of transfers, with clubs with positive balances being once again associated with higher odds.

These insights do not change if we add the variables regarding team age and attendances. The statistics associated with the quality of the estimates allow us to observe that, among those shown in Table 3, the estimation that includes the points of the teams under negotiation, the respective differences in terms of points at the final standings, the value of transfers and the difference in the balance of transfers, for beyond the age of the team and the level of the attendances is the estimation with the highest values of Log-likelihood, Pseudo-R2, and Correctly classified (%).

## 4.3 Identifying determinants for the values of the transfers observed

Taking advantage of the constructed database, we extended the main effort of this work, which was to test determinants for the transfer flows of soccer players in the Spanish League in the most expensive seasons. Thus, we extended the previous effort to try to find determinants for the values of the transfers observed. Our dependent variable is thus, for this model, the value of transfers from club A to club B at the observed time. The independent variables are the variables already tested in the previous empirical steps (Tables 2 and 3).

To estimate the model relating to the value of transfers between club A and a different destination, and as we are dealing with panel data, we used Ordinary Least Squares with Random Effects (as explained in the methodology section). In a generalized interpretation of the values in Table 4, we conclude that larger point differences, as well as the other analyzed mismatches (age of the team, balances of transfers and attendances along the seasons) increase the value of the observed transfers. There is also a positive effect on the part of the acquiring teams if these acquiring teams have higher values for the variables of age, points achieved, transfer balances and attendances.

Let us now discuss these results. As mentioned, with this work we analyzed, in great detail, the seasons of the Spanish championship that had the highest values. We wanted to understand not only the determinants responsible for the structure of the transfer network but also the values involved. We remember that previous works had focused on the importance of studying the network structure for a modern analysis of the phenomenon of professional transfers

**Table 4. Panel LS random effects estimates for La Liga players' transfers (2018–2019 and 2019–2020).**

|  | RE estimates | Marginal effects |
|---|---|---|
| Points16_Difference | 0.234*** (0.084) | 0.634*** (0.033) |
| Points16_Buying team | 0.516* (0.262) | 1.002*** (0.371) |
| Points17_Difference | 0.512** (0.187) | 0.987*** (0.217) |
| Points17_Buying team | 0.545*** (0.167) | 1.089*** (0.228) |
| Points18_Difference | -0.636 (0.369) | -1.267 (0.769) |
| Points18_Buying team | 0.472** (0.197) | 0.877** (0.151) |
| TrfBalan16_Difference | 0.478* (0.168) | 0.489** (0.188) |
| Trfbalanc16_Buying team | 0.399*** (0.121) | 0.729* (0.349) |
| TrfBalan17_Difference | 0.221* (0.121) | 0.171*** (0.041) |
| TrfBalan17_Buying team | 0.152*** (0.022) | 0.363* (0.233) |
| TrfBalan18_Difference | 0.395*** (0.184) | 0.954*** (0.154) |
| TrfBalan18_Buying team | 0.215*** (0.071) | 0.435** (0.155) |
| Seasons dummies | No |  |
| N | 882 |  |
| R2 (Overall) | 0.554 |  |
| Points16_Difference | 0.233*** (0.083) | 0.633*** (0.030) |
| Points16_Buying team | 0.514* (0.242) | 1.042** (0.471) |
| Points17_Difference | 0.552** (0.185) | 0.957*** (0.215) |
| Points17_Buying team | 0.546*** (0.166) | 1.086*** (0.226) |
| Points18_Difference | -0.676 (0.367) | -1.265 (0.765) |
| Points18_Buying team | 0.478** (0.198) | 0.879** (0.159) |
| TrfBalan16_Difference | 0.408** (0.108) | 0.449** (0.184) |
| Trfbalanc16_Buying team | 0.395*** (0.125) | 0.725* (0.345) |
| TrfBalan17_Difference | 0.228* (0.128) | 0.174*** (0.044) |
| TrfBalan17_Buying team | 0.159*** (0.029) | 0.360* (0.203) |
| TrfBalan18_Difference | 0.394*** (0.184) | 0.954*** (0.144) |
| TrfBalan18_Buying team | 0.213*** (0.073) | 0.434** (0.154) |
| Team_age_Difference | 0.487* (0.223) | 0.832* (0.333) |
| Team_age_Buying team | 0.577** (0.201) | 0.932** (0.462) |
| Seasons dummies | No |  |
| N | 882 |  |
| R2 (Overall) | 0.574 |  |
| Points16_Difference | 0.238*** (0.088) | 0.683 (0.830) |
| Points16_Buying team | 0.594* (0.249) | 1.942** (0.479) |
| Points17_Difference | 0.502** (0.180) | 0.950*** (0.210) |
| Points17_Buying team | 0.541*** (0.161) | 1.186*** (0.216) |
| Points18_Difference | -0.626 (0.362) | -1.225 (0.762) |
| Points18_Buying team | 0.473** (0.193) | 0.839** (0.139) |
| TrfBalan16_Difference | 0.448** (0.104) | 0.444** (0.144) |
| Trfbalanc16_Buying team | 0.355*** (0.155) | 0.755* (0.355) |
| TrfBalan17_Difference | 0.226* (0.126) | 0.176*** (0.044) |
| TrfBalan17_Buying team | 0.179*** (0.027) | 0.367** (0.273) |
| TrfBalan18_Difference | 0.398*** (0.188) | 0.958*** (0.148) |
| TrfBalan18_Buying team | 0.219** (0.093) | 0.439** (0.159) |
| Team_age_Difference | 0.480** (0.203) | 0.830* (0.330) |
| Team_age_Buying team | 0.571** (0.201) | 0.932** (0.412) |
| Assistance_Difference | 0.409* (0.201) | 0.800*** (0.203) |

*(Continued)*

**Table 4.** (Continued)

| | RE estimates | Marginal effects |
|---|---|---|
| Assistance_Buying team | 0.311* (0.151) | 0.602* (0.119) |
| Seasons dummies | No | |
| N | 882 | |
| R2 (Overall) | 0.614 | |

Robust standard errors between parentheses. Significance level: 1%, ***; 5%, **; 10%, *.

between sports clubs [18–21]. Other works had focused on some dimensions of footballer transfers in Spain [1]. However, our work has shown how analyzing the network structure is essential to understanding the phenomenon under analysis, especially the differentiation of values involved.

We emphasize that these results converge with some authors but advance the current state of the art. They converge with [2, 4, 18] in the sense that there is a strategic behavior of clubs in transfers. Players are transferred as an important source of revenue for the club's financial cycle, but they are also transferred so that the expected sporting objectives are guaranteed. This strategic behavior of oligopolistic agents had already been detected in previous works: [2, 4, 18]. However, our study went further–it demonstrated that the greater the differences between the values of a financial or sporting variable observed for two teams, the greater the probability of a player transfer between these two teams in the Spanish championship. We also observed that the greater the differences between these teams (in financial or sporting variables) the greater the values involved in the observed transfers.

These results surpass existing literature as in [5]. Specifically, our results have important implications–the use of transfers as an instrument for the strategic management of clubs cannot be neglected as an important cause for existing imbalances, particularly in terms of competitive balance. Secondly, the observed flow could compromise the financial and sporting viability of smaller teams, which are much more dependent on the surplus resources of larger teams. Thirdly, the structure of the network designed by transfers clearly shows the dependence of the same network on a small group of clubs. This dependence is a serious threat to the integrated objectives of the entire industry, requiring reflection in favour of a more sustainable structure.

## 5. Conclusion, implications and challenges

This work studies in depth soccer transfers in the Spanish League in the most expensive seasons of its History (2018/2019 and 2019/2020 seasons). Following the literature on the topic, we have tested the hypothesis that dominant teams create hierarchical structures in which they move more players to lower-ranked teams rather than to other highly ranked teams. Also, we identify determinants for the values of the transfers observed.

This work confirmed, through the originality of Network Analysis, the existence of an oligopolistic structure with strategic behavior in the professional football industry in Spain. Additionally, this work showed that transfers are a vital instrument of this strategy between oligopolists. However, we want to emphasize that this instrument is aimed at temporal consistency, reducing the possibility of mid-table teams being champion teams in Spain. This type of consequence is important for the discussion of the lack of sporting competitiveness in the championship, reduced to a competition whose biggest prize is only within the reach of a small group of teams.

The review of the previous literature identified works which studied dimensions responsible for the value of transfers in the world of professional soccer. However, the approach of such intense dynamics between clubs has not yet been widely discussed using Network Analysis. In this way, we tested four models. We started with only one parameter and the results suggested that the existence of ties in the observed network is less likely to be explained by chance. Then, we explored a second model in which we included structural parameters (there we added reciprocity). Also, we included the possibility of certain attributes explaining the properties of the network in a third model. Finally, we have tested combined model with structural parameters and attributes. At last, we have been able to provide a more complete perspective of situation. In this way, we tested a larger set of characteristics involving each element of the dyad of teams involved in a transfer process for our Hypothesis. Considering this hypothesis, Spanish teams tend to transfer players with teams that are not competing for the same rank, which is a behavior fitting markets. Additionally, we used Ordinary Least Squares with Random Effects to identify determinants for the values of the transfers observed. We concluded that larger point differences, age of the team, balances of transfers and assists throughout the seasons increase the value of the transfers observed.

We recognize two major implications from this empirical effort. The first implication is the identification of the Spanish Soccer League with an oligopolist market which deserves additional study to preserve and maintain the competitive balance, a structural factor for the financial sustainability of professional sports. Additionally, the non-randomness of the transfer process implies that resources such as information on players and teams or the role of players' agents/'third-parties' are elements that are able to generate rents, which can be associated with an increasing inequality among teams and with a threat to the already mentioned competitive balance in professional sports.

These results stimulate further works. First, we suggest investigating not only the two most recent seasons but also a longer period and for other professional soccer leagues. This would enhance the robustness of these original achievements. Additionally, we also suggest enlarging the observed and tested attributes explaining the reported transfers, i.e., enlarging the analysis to other independent variables than those tested here for the exponential random graph models. We also consider relevant the possibility of detailing these results by considering the continents of the leagues which were the origins or the destinations of the observed players' transfers. Finally, given the structure of time series easily perceived for the evolution of the indicators analyzed for the weekly and monthly networks of transfers, we also consider running a structural breaks test on these relevant series, as well testing Markov-Switching models for deepening the study of these observations.

## Supporting information

**S1 Data.**
(XLSX)

## Author Contributions

**Conceptualization:** Jesyca Salgado-Barandela.

**Data curation:** Jesyca Salgado-Barandela.

**Formal analysis:** Jesyca Salgado-Barandela.

**Funding acquisition:** Paulo Reis Mourao.

**Investigation:** Paulo Reis Mourao.

**Methodology:** Paulo Reis Mourao.

**Resources:** Jesyca Salgado-Barandela.

**Software:** Paulo Reis Mourao.

**Supervision:** Paulo Reis Mourao.

**Validation:** Paulo Reis Mourao.

**Visualization:** Jesyca Salgado-Barandela.

**Writing – original draft:** Jesyca Salgado-Barandela.

**Writing – review & editing:** Paulo Reis Mourao, Jesyca Salgado-Barandela.

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
