## [Decision Letter · Decision Letter 0]

12 Sep 2023

PONE-D-23-16261Exploring Soccer transfers in Spanish League: the hidden role of strategic differences among teamsPLOS ONE

Dear Dr. Salgado-Barandela,

Thank you for submitting your manuscript to PLOS ONE. After careful consideration, we feel that it has merit but does not fully meet PLOS ONE’s publication criteria as it currently stands. Therefore, we invite you to submit a revised version of the manuscript that addresses the points raised during the review process.

ACADEMIC EDITOR:  Thank you for submitting your manuscript, "Exploring Soccer Transfers in the Spanish League: The Hidden Role of Strategic Differences Among Teams," to our international journal. I appreciate the effort you've put into this research and would like to guide you on addressing the comments and concerns raised by our reviewer.

Overall Impression: The reviewer raised several valid points that need to be addressed to enhance the quality and clarity of your manuscript. Here are the main issues and recommendations for improvement:

1. Knowledge Gap and Theoretical Foundation:

The introduction should establish the knowledge gap that your study addresses. What specific contribution does your research make to the existing literature?Please strengthen the theoretical foundation in the introduction to give readers a better understanding of the context and significance of your study.

2. Theoretical Support for Hypotheses:

Ensure that your hypotheses are well-grounded in relevant theories or existing literature. Provide a clear theoretical framework to support your research questions.

3. Data and Sample Selection:

Clarify the data collection period and its relevance to the study. Address the discrepancy in the data collection period and how it relates to the 2017-18 and 2018-19 seasons.Explain why you chose these particular seasons, especially if they are considered outliers. Consider discussing how these seasons contribute to the uniqueness of your study.

4. Contribution and Research Focus:

Could you clearly explain the specific contributions of your study? What new insights or knowledge does it bring to soccer transfers?Address the reviewer's concerns about the known nature of certain transfer patterns and consider expanding your research to include factors like transfers from B-teams, as suggested.

5. Selection of Variables:

Provide a thorough justification for your choice of variables. Consider incorporating sporting variables, such as playing style, to enhance the comprehensiveness of your analysis.I'd like you to reference relevant studies, like the one mentioned by the reviewer, to support your variable selection.

6. International Transfers:

Discuss international transfers and their significance in the context of your study. Explain why your analysis did not include them, or consider including them if appropriate.

7. Discussion Section:

Develop a robust discussion section that goes beyond presenting results. Engage with relevant literature and discuss the implications of your findings in depth. This section should include more than just recapitulate results but provide a critical analysis.

8. Language Editing:

Prioritize language editing to ensure clarity and readability throughout the manuscript. Clear, concise language is crucial for effective communication.

Way Forward:

Focus your research on exploring soccer transfer networks and consider including transfers from B-teams and international transfers, as suggested by the reviewer.Strengthen the theoretical framework and clearly explain the knowledge gap your study fills.Ensure thorough justification for variable selection and incorporate relevant sporting variables.Enhance the discussion section by critically engaging with the results and their implications.

Please make the necessary revisions and provide a detailed response to each of these concerns in your revised manuscript. Addressing these issues will significantly improve the quality and impact of your research. We look forward to receiving your revised manuscript.

We look forward to receiving your revised manuscript.

Kind regards,

Ricardo Limongi

Academic Editor

PLOS ONE

Reviewers' comments:

Reviewer's Responses to Questions

**Comments to the Author**

1. Is the manuscript technically sound, and do the data support the conclusions?

Reviewer #1: Partly

Reviewer #2: Partly

2. Has the statistical analysis been performed appropriately and rigorously? 

Reviewer #1: Yes

Reviewer #2: No

3. Have the authors made all data underlying the findings in their manuscript fully available?

Reviewer #1: Yes

Reviewer #2: Yes

4. Is the manuscript presented in an intelligible fashion and written in standard English?

Reviewer #1: Yes

Reviewer #2: No

5. Review Comments to the Author

Reviewer #1: Dear Editor,

I appreciate the opportunity to review the manuscript Exploring Soccer transfers in Spanish League: the hidden role of strategic differences among teams. The aim of this manuscript was “test determinants for the transfer relationships of football players in the Spanish League in the most expensive seasons with records (2017/2018 and 2018/2019 seasons).”. After reading the manuscript, it was possible to observe that:

1 – It is not clear what knowledge gap will be filled.

2 – There is little theoretical foundation in the introduction.

3 – There is no theoretical support for the hypotheses.

4 – The study sample is old and does not make sense today (more than 4 years have passed).

5 – There is no discussion in the article. In the “Empirical analysis” section, a more in-depth discussion with other authors should have taken place, which did not happen. It seems to me that this section is a continuation of the results.

Given the above, I suggest rejecting the manuscript.

Reviewer.

Reviewer #2: Report on PONE-D-23-16261

Title: Exploring Soccer transfers in Spanish League: the hidden role of strategic differences among teams

Thank you for giving me the opportunity to read the paper.

The authors studied players transfers in Spanish football league (La Liga) for the seasons 2017-18 and 2018-19, using exponential random graph models (ERGM). While I agree that there is some novelty in application of ERGM and identification of the role of networks in football transfer, I could not understand what the focus of the paper is.

First I will highlight the major issues with the paper and then talk about some minor issues.

Major issues

1. Data

The authors took data for 63 weeks between 1st July 2018 and 1st September 2019. In Europe, there are two transfer windows – one occurs at the beginning of the season (known as the summer window) and another at the mid-season (known as the winter window). The given period covers both the transfer windows of 2018-19 season and the summer window of 2019-20 season. I am unable to understand how the authors claim to analyze the transfers for 2017-18 and 2018-19 seasons using that data. The authors must clarify this.

Moreover, according to the authors, the selected seasons are the most expensive two seasons. Doesn’t that make these seasons outliers? Can anything be generalized based on outlier data?

2. What did we learn?

It is not clear what the contribution of the paper is. It is well-known that a few clubs dominate all the major football leagues in Europe. The oligopolistic structure exists outside of Spain too – in Germany, France, and England (and to a large extent in Italy). The fact that such an oligopolistic structure distorts the competitive balance is also a well-researched topic.

Identification of the fact that teams tend to transfer players with teams that don’t compete for the same spot (in the league table) is also a known fact in football. Typically, in most leagues, rookie players who do well in low-ranked teams are picked up by high-ranked teams.

The story of transfer network also involves the B-teams. In Spain, big clubs have their B-teams (and also academies) that play in the second division. Players from these B-teams are often transferred to the mid-ranked teams of the first division. The authors may include such transfers to make the story behind their empirical analysis more cohesive.

3. Selection of variables

The authors did not justify the selection of variables.

More importantly, they did not consider any sporting variable. Team playing style may be an important factor. For example, the authors may refer to:

McHale, I. G., & Holmes, B. (2023). Estimating transfer fees of professional footballers using advanced performance metrics and machine learning. European Journal of Operational Research, 306(1), 389-399.

4. International transfer

Transfer balance includes the amount received and spent on international transfers, which is a significant part of transfers taking place in each season. Many South American players come to Spain from South American clubs. How can the authors ignore that?

Minor issue

The paper requires language editing.

Way forward

1. Focus on transfer networks

2. Include transfers from B-teams.

3. Include international transfers (or exclude the amount received from and spent on international transfers, which may be challenging).

4. Include sporting variables like playing style.

6. PLOS authors have the option to publish the peer review history of their article (what does this mean?). If published, this will include your full peer review and any attached files.

Reviewer #1: No

Reviewer #2: **Yes: **Sumit Sarkar

---

## [Author Response · Author response to Decision Letter 0]

24 Oct 2023

We have considered the comments of both reviewers and academic editor, responding point by point to each of the recommendations. 

ACADEMIC EDITOR: 

1. Knowledge Gap and Theoretical Foundation:

- The introduction should establish the knowledge gap that your study addresses. What specific contribution does your research make to the existing literature?

Thanks for the suggestion. We have clarified the knowledge gap in introduction section:

The main knowledge gap covered by this work is to study the dimensions responsible for the value of transfers from an approach of intense dynamics between clubs using network analysis. This work adds to the high-potential area of social network analysis in sports organizations [1,9,12,16]. In this way, empirical analysis shows relevant findings. First, Spanish soccer transfers are not random, and this evidence reflects the environment of imperfect competition characterizing La Liga. Second, the non-randomness of the transfer process can be associated with an increasing inequality among teams and with a threat to the competitive balance in professional sports. Additionally, we identify that larger point differences, as well as age of the team, balances of transfers and attendances throughout the seasons increase the value of the transfers observed.

- Please strengthen the theoretical foundation in the introduction to give readers a better understanding of the context and significance of your study.

Thanks for the recommendation. We have strengthened the theorical foundation in the introduction. We have expanded the explanation of the importance of the study, basing it on existing literature:

“When a player moves from one club to another, his registration is transferred to that club, normally for a fee agreed between the two parties. This is the definition of football transfer, which involves two aspects: the transfer of the player from one team to another and the amount of money agreed between both teams. Transfers in the football world have become a hot topic in academic studies in recent years. The monetary resources involved as well as the growing trend helped in this highlight. The works focused on soccer transfers allow the identification of several lines of development. With no pretensions of exhaustion, we highlight 3 common lines in terms of publications. In a first line, there is the discussion of determinants for such flows, therefore determinants for transfers, either in number of players involved or in the associated amount [1,2,3,4,5,6,7]. In a second line, there are works like Mourao [2,8] that seek to assess the efficiency of transfers considering their own restrictions on the part of the involved determinants. Finally, in a third line, the specificity of the transfers is evaluated in the consideration of focuses, such as the places occupied by some athletes, the nature of the national championships involved or the relationship with certain observed periods [9,10,11].

In a special way, the study of the determinants of transfers has achieved an important dimension in professional soccer. As Mourao [2] explains, the provisions of Financial fair play have pushed clubs to adopt transfer market strategies to save on relevant expenses and amortize transfers. At the same time, this strategy has increased the level of relevant income in terms of capital gains on player disposals and this effect is stronger for clubs from professional soccer leagues [12]. On the other hand, there are professional soccer leagues that show a notable increase in the monetary resources of player´s transfers. This situation represents an opportunity to analyse the factors that influence transfers. Furthermore, scientific studies have evolved towards the consideration of new statistical tools in the study of determinants in this topic, such as network analysis [13].”

2. Theoretical Support for Hypotheses:

- Ensure that your hypotheses are well-grounded in relevant theories or existing literature. Provide a clear theoretical framework to support your research questions.

Thanks for the suggestion. We have rewritten the literature review section to provide a clear theoretical framework to support your research questions:

“Previous literature has investigated the main drivers of transfer fees paid by clubs. The evidences indicate that, the level of a transfer fee is linked to the player’s characteristics and performance features [6,7,14, 15]. Franceschi

 et al. [16] identifies in his systematic review study that age and age squared are the most often tested variables, consistently associated with significant. In addition to age, variables such as usage of the player (minutes played or number of appearances) and international status and decisiveness (goals and assists) are frequently included in the studies and positively and consistently influence the valuation of the football player. On the other hand, Franceschi et al. [16] explains that this positive relationship can be explained by the low-score nature of football. When the usage variables are studied together with the performance variables, the significant results are low. Furthermore, these indicators are probably correlated, thus they may weaken each other’s influence on the dependent variable. 

Previous studies also consider the economic-financial dimension in the analyzed determinants. In a specific way, economic variables are defined such as wage rate, relevant incentives, and the remaining duration of the contract, play a role in determining the transfer fee for player registration [5,17]. In relation to the financial dimension, variables such as the values of past transfers and the respective balances, in addition to the financial indicators observed from the Account Reports of the Clubs [2,7]. Campa [5] explains that they reflect the influence of factors related to neither players’ skills nor to other economic features, for example: the race of a player or the number of Google searches associated with the player. 

In relation to the dependent variable, most of the works in extant literature propose at least one model specification where the dependent variable is the transfer fee. It is necessary to consider that transfer fee it is not available for every player at any time but rather only when a transfer eventually occurs [12,16]. For this reason, studies of transfer determinants usually use crowdsourced estimates of players value, retrieved from the website Transfermarkt [16]. It is possible to find differences in relation to the use of transfer fee as a dependent variable. Matesanz et al. [18] use three variables to construct transfer data: transfer spending (transfer spending is the amount of money each club has spent in transfers in each season), transfer earnings (represent the amount of money each club has generated from transfers in each season) and transfer fee volume (transfer spending + transfer earnings). On the other hand, Liu et al. [19] and Rosseti and Caproni [20] detail transfers by type of transaction: free / payed transfer, and loans. In the case of Coates et al. [21] they treat loans also as a connection between clubs.

In the literature, only a small number of papers have analyzed football transfers using network analysis. Liu et al. [19] studied the relationship between the success of a club and player transfers on a dataset of transfer records from 2011 to 2015 of 410 professional clubs in 24 world-wide top class leagues. They show “that clubs’ match performance and profitability from the transfer market are strongly associated with the coreness and brokerage properties of their corresponding nodes in the player transfer network” [19]. Moreover, Matesanz et al. [18] explore the evolution of the football players’ transfer network among 21 European first leagues between the seasons 1996/1997 and 2015/2016. They get findings similar to Liu et al. [19]. Rosseti and Caproni [20] also studied the impact of club transfer strategies on team performance, but with a different database. In this case, they worked with data covering 25 years of trading among clubs of the 6 FIFA federations. Unlike Liu et al. [19], Rosseti and Caproni [20] found support for a nonlinear relationship between network characteristics and roster changes. 

On the other hand, Coates et al. [21] extend the work previously done using a massive dataset on transfers and club characteristics to study the relation between transfer strategy and performance. They use data on 23220 clubs from 189 countries from 1996 through 2016 and they found that a wide transfer network is harmful for a club’s sport performance, and the best strategy is to have the lowest number of connections inside the league. As Coates et al. [21] explains “for financial performance, it is better to have a high number of international connections with clubs that have widespread transfer networks, whereas connections inside domestic league are not important”.

More recently, Clemente y Cornaro [13] have applied network analysis to study number of transfers and the related costs for football players in the world market. They use all transfers between football clubs regarding the season 2020–2021, taking into account both winter and summer sessions, 44,481 transfers in total. They find that clubs of top football leagues tend to behave in a similar way in the network and to trade players each other and highlighted the role of several European countries that represent top leagues in the soccer market and are also involved as central countries in the economic trades.

In relation to those studies [13,18,19,20,21] it is identified that the number of points is a variable commonly used as an independent or control variable. Liu et al. [19] use Kendall’s Tau coefficient to analyze the relationship between club functionalities versus network properties in international and domestic transfer networks, node properties in transfer network and loan network, and node properties in money leagues and farm leagues. These authors consider two variables in relation to the number of points, which are: average league points (domestic league) and IFFHS Club World Ranking points (The IFFHS Club World Ranking points are composed by the International Federation of Football History & Statistics (IFFHS) based on a set of rules composing both domestic and international match results). While, Matesanz et al. [18] include domestic rank and UEFA points to study the football player transfer market activities among European first leagues from 21 countries. Rosseti and Caproni [20] and Coates et al. [21] only consider points per game like a dimension to represent number of points. 

In this same line, previous papers also consider characteristics of the players in the selection of independent or control variables. Matesanz et al. [18] and Rosseti and Caproni [20] include various characteristics in relation to the transferred players. In the case of Matesanz et al. [18] variables are: age, nationality, field position. Matesanz et al. [18] find that “the connection between transfer spending and sportive performance, especially in UEFA competitions, is extremely strong particularly for clubs from the top cluster, which might further limit the overall level of uncertainty of outcome”. While, Rosseti and Caproni [20] include role, age and market value to players transferred, and they identify among other findings that roster stability plays a crucial role on the obtained results and more you change more unpredictable your future results will be. Coates et al. [21] include player´s rating in its methodological model. Also, it is the only work that also measures the performance of the coach using fixed effects for its measurement. Finally, the attendance variable is only included by Coates et al. [21], the rest of previous works do not consider it in its measurement.”

3. Data and Sample Selection:

- Clarify the data collection period and its relevance to the study. Address the discrepancy in the data collection period and how it relates to the 2017-18 and 2018-19 seasons.

We would like to clarify that we have made a mistake when writing the analysis period. The true analysis period is 2018-2019 and summer window of 2019-2020 seasons. 

- Explain why you chose these particular seasons, especially if they are considered outliers. Consider discussing how these seasons contribute to the uniqueness of your study.

The chosen seasons are the most expensive but cannot be considered outliers. Rather, this situation responds to an upward tendency in transfers in professional soccer leagues. We have strengthened the theorical foundation in the introduction to explain why the most expensive seasons have been selected. We consider that it is important to understand what happens in these seasons because this provides relevant findings to explain a scenario of upward tendency in transfers. Situation that not only occurs in the Spanish league.

We explain this in data section:

“Thus, three reasons emerge for us to focus our study on these times, without neglecting to extend to more remote times. These three reasons are: the maximum achieved in relation to sports transfers in Spain, the emerging role of network analysis to study competitive phenomena, and the importance of observing the dimensions of oligopolies that exist in sport. First, the maximums observed during the transfer windows listed here, due to the current pandemic, have not yet been exceeded in the most recent transfer windows. It is important to understand what happens in these seasons because this provides relevant findings to explain a scenario of upward tendency in transfers. This situation is not exclusive to the Spanish Soccer League; other leagues also have high transfer costs. Thus, the understanding of a general situation of the transfer market can be improved.

Second, the transfer market has not considered so far either the role of the networks involved or the importance of more likely transfers due to imbalances in characteristics between clubs. Thirdly, behaviors induced in oligopolistic environments, such as that observed in professional soccer championships, induce asset transfers between companies that compete for different market shares, a central hypothesis in our study.

Our dependent variable for the model proposed in this study is the value of transfers from club A to club B at the observed seasons. The independent variables are the variables shown in Table 1, which follow the quoted literature from Section 2. In addition to the variables in Table 1, certain attributes are added to the model to explain the properties of the network. These attributes are: number of points obtained at the end of season, transfer balance value (millions of euros), average age of the team squad for the seasons and total attendance at the team’s stadium for the seasons.”

4. Contribution and Research Focus:

- Could you clearly explain the specific contributions of your study? What new insights or knowledge does it bring to soccer transfers?

Thanks for the suggestion. We have clarified the knowledge gap in introduction section:

The main knowledge gap covered by this work is to study the dimensions responsible for the value of transfers from an approach of intense dynamics between clubs using network analysis. This work adds to the high-potential area of social network analysis in sports organizations [1,9,13,17]. In this way, empirical analysis shows relevant findings. First, Spanish soccer transfers are not random, and this evidence reflects the environment of imperfect competition characterizing La Liga. Second, the non-randomness of the transfer process can be associated with an increasing inequality among teams and with a threat to the competitive balance in professional sports. Additionally, we identify that larger point differences, as well as age of the team, balances of transfers and attendances throughout the seasons increase the value of the transfers observed.

- Address the reviewer's concerns about the known nature of certain transfer patterns and consider expanding your research to include factors like transfers from B-teams, as suggested.

The main conclusions are now explained in a clearer way.

This work confirmed, through the originality of Network Analysis, the existence of an oligopolistic structure with strategic behavior in the professional football industry in Spain. Additionally, this work showed that transfers are a vital instrument of this strategy between oligopolists. However, we want to emphasize that this instrument is aimed at temporal consistency, reducing the possibility of mid-table teams being champion teams in Spain. This type of consequence is important for the discussion of the lack of sporting competitiveness in the championship, reduced to a competition whose biggest prize is only within the reach of a small group of teams. A different paper would be relevant for a further study only highlighting the B-teams. 

5. Selection of Variables:

- Provide a thorough justification for your choice of variables. Consider incorporating sporting variables, such as playing style, to enhance the comprehensiveness of your analysis. I'd like you to reference relevant studies, like the one mentioned by the reviewer, to support your variable selection.

We have rewritten the literature review section to provide a clear theoretical framework to provide a thorough justification for our choice of variables. Several papers have been read and quoted after the suggestion of the reviewers, including the very good references provided by the reviewers: 

“Previous literature has investigated the main drivers of transfer fees paid by clubs. The evidences indicate that, the level of a transfer fee is linked to the player’s characteristics and performance features [6,7,14, 15]. Franceschi et al. [16] identifies in his systematic review study that age and age squared are the most often tested variables, consistently associated with significant. In addition to age, variables such as usage of the player (minutes played or number of appearances) and international status and decisiveness (goals and assists) are frequently included in the studies and positively and consistently influence the valuation of the football player. On the other hand, Franceschi et al. [16] explains that this positive relationship can be explained by the low-score nature of football. When the usage variables are studied together with the performance variables, the significant results are low. Furthermore, these indicators are probably correlated, thus they may weaken each other’s influence on the dependent variable. 

Previous studies also consider the economic-financial dimension in the analyzed determinants. In a specific way, economic variables are defined such as wage rate, relevant incentives, and the remaining duration of the contract, play a role in determining the transfer fee for player registration [5,17]. In relation to the financial dimension, variables such as the values of past transfers and the respective balances, in addition to the financial indicators observed from the Account Reports of the Clubs [2,7]. Campa [5] explains that they reflect the influence of factors related to neither players’ skills nor to other economic features, for example: the race of a player or the number of Google searches associated with the player. 

In relation to the dependent variable, most of the works in extant literature propose at least one model specification where the dependent variable is the transfer fee. It is necessary to consider that transfer fee it is not available for every player at any time but rather only when a transfer eventually occurs [12,16]. For this reason, studies of transfer determinants usually use crowdsourced estimates of players value, retrieved from the website Transfermarkt [16]. It is possible to find differences in relation to the use of transfer fee as a dependent variable. Matesanz et al. [18] use three variables to construct transfer data: transfer spending (transfer spending is the amount of money each club has spent in transfers in each season), transfer earnings (represent the amount of money each club has generated from transfers in each season) and transfer fee volume (transfer spending + transfer earnings). On the other hand, Liu et al. [19] and Rosseti and Caproni [20] detail transfers by type of transaction: free / payed transfer, and loans. In the case of Coates et al. [21] they treat loans also as a connection between clubs.

In the literature, only a small number of papers have analyzed football transfers using network analysis. Liu et al. [19] studied the relationship between the success of a club and player transfers on a dataset of transfer records from 2011 to 2015 of 410 professional clubs in 24 world-wide top class leagues. They show “that clubs’ match performance and profitability from the transfer market are strongly associated with the coreness and brokerage properties of their corresponding nodes in the player transfer network” [19]. Moreover, Matesanz et al. [18] explore the evolution of the football players’ transfer network among 21 European first leagues between the seasons 1996/1997 and 2015/2016. They get findings similar to Liu et al. [19]. Rosseti and Caproni [20] also studied the impact of club transfer strategies on team performance, but with a different database. In this case, they worked with data covering 25 years of trading among clubs of the 6 FIFA federations. Unlike Liu et al. [19], Rosseti and Caproni [20] found support for a nonlinear relationship between network characteristics and roster changes. 

On the other hand, Coates et al. [21] extend the work previously done using a massive dataset on transfers and club characteristics to study the relation between transfer strategy and performance. They use data on 23220 clubs from 189 countries from 1996 through 2016 and they found that a wide transfer network is harmful for a club’s sport performance, and the best strategy is to have the lowest number of connections inside the league. As Coates et al. [21] explains “for financial performance, it is better to have a high number of international connections with clubs that have widespread transfer networks, whereas connections inside domestic league are not important”.

More recently, Clemente y Cornaro [13] have applied network analysis to study number of transfers and the related costs for football players in the world market. They use all transfers between football clubs regarding the season 2020–2021, taking into account both winter and summer sessions, 44,481 transfers in total. They find that clubs of top football leagues tend to behave in a similar way in the network and to trade players each other and highlighted the role of several European countries that represent top leagues in the soccer market and are also involved as central countries in the economic trades.

In relation to those studies [13,18,19,20,21] it is identified that the number of points is a variable commonly used as an independent or control variable. Liu et al. [19] use Kendall’s Tau coefficient to analyze the relationship between club functionalities versus network properties in international and domestic transfer networks, node properties in transfer network and loan network, and node properties in money leagues and farm leagues. These authors consider two variables in relation to the number of points, which are: average league points (domestic league) and IFFHS Club World Ranking points (The IFFHS Club World Ranking points are composed by the International Federation of Football History & Statistics (IFFHS) based on a set of rules composing both domestic and international match results). While, Matesanz et al. [18] include domestic rank and UEFA points to study the football player transfer market activities among European first leagues from 21 countries. Rosseti and Caproni [20] and Coates et al. [21] only consider points per game like a dimension to represent number of points. 

In this same line, previous papers also consider characteristics of the players in the selection of independent or control variables. Matesanz et al. [18] and Rosseti and Caproni [20] include various characteristics in relation to the transferred players. In the case of Matesanz et al. [18] variables are: age, nationality, field position. Matesanz et al. [18] find that “the connection between transfer spending and sportive performance, especially in UEFA competitions, is extremely strong particularly for clubs from the top cluster, which might further limit the overall level of uncertainty of outcome”. While, Rosseti and Caproni [20] include role, age and market value to players transferred, and they identify among other findings that roster stability plays a crucial role on the obtained results and more you change more unpredictable your future results will be. Coates et al. [21] include player´s rating in its methodological model. Also, it is the only work that also measures the performance of the coach using fixed effects for its measurement. Finally, the attendance variable is only included by Coates et al. [21], the rest of previous works do not consider it in its measurement.”

Also, we recall the underlying reason as well as the supporting literature below in data section. 

“The number of points in previous seasons is a variable associated with the value of the team's budget as well as the ability to generate transfers around the club. The use of values from several epochs allows for a more robust associated reading [18, 19]. The average value of the number of points allows you to control possible fluctuations outside the predictable performance of each team.

The variable focused on the transfer balance is a variable that shows the difference between the amount received from transfers made and the amount paid due to transfers that entered the club. In line with the literature consulted [18], higher transfer balances produce a leverage effect on the club's future operations. Once again, the presence of a variable associated with the average value (in this case, the balance of transfers) aims to provide greater robustness to the model.

The average age of the team shows the need to renew the squad in future exercises. As suggested by Samur [22], aging teams increase the probability of incoming transfers in the future.

The total number of spectators reported for each team's home games is a variable that controls the number of points as well as the balance of transfers. Teams with good home game attendances show significant support from the surrounding community. The inclusion of the average attendance at home aims to control possible imbalances that consecutive fluctuations in attendance may suffer [23].”

In relation to incorporating sporting variables, Franceschi, et al. (2023) explain that usage of the player (minutes played or number of appearances), international status and decisiveness (goals and assists) positively and consistently influence the valuation of the football player. On the other hand, Franceschi, et al. (2023) find that the low-score nature of football can explain why the decisiveness of the players estimated through their ability to score and to assist is positively linked with their valuation. Even if the number of specifications in which the performance variables (goals, assists, matches, minutes played, cards) are found to be significant can seem low, these indicators are often tested conjointly even if they are probably correlated, thus they may weaken each other’s influence on the dependent variable. However, we consider it relevant to consider some sporting variables in additional studies.

6. International Transfers:

- Discuss international transfers and their significance in the context of your study. Explain why your analysis did not include them or consider including them if appropriate. Transfer balance includes the amount received and spent on international transfers, which is a significant part of transfers taking place in each season. Many South American players come to Spain from South American clubs. How can the authors ignore that?

Many players have been transferred to the Spanish championship (as well as other European championships) from the various South American championships. However, the weight of these transfers, both in number and volume, is small compared to the number and total volume of registered transfers. Furthermore, we included in our analysis, despite the previous observation, these transfers from Latin America, which are (like the others) in the “Others” Group (see Figure 1). We also consider that a potential alternative work would involve discriminating the results explained here based on the continents of origin or destination of the observed transfers (please also check our paragraphs regarding further challenges for research).

7. Discussion Section:

- Develop a robust discussion section that goes beyond presenting results. Engage with relevant literature and discuss the implications of your findings in depth. This section should include more than just recapitulate results but provide a critical analysis.

Now, there is a substantial extension of the empirical discussion, also relating the evidence with the previous literature (page 25-26): 

“Let us now discuss these results. As mentioned, with this work we analyzed, in great detail, the seasons of the Spanish championship that had the highest values. We wanted to understand not only the determinants responsible for the structure of the transfer network but also the values involved. We remember that previous works had focused on the importance of studying the network structure for a modern analysis of the phenomenon of professional transfers between sports clubs [18, 19, 20, 21]. Other works had focused on some dimensions of footballer transfers in Spain [1]. However, our work has shown how analyzing the network structure is essential to understanding the phenomenon under analysis, especially the differentiation of values involved.

We emphasize that these results converge with some authors but advance the current state of the art. They converge with [2], [4], [18] in the sense that there is a strategic behavior of clubs in transfers. Players are transferred as an important source of revenue for the club's financial cycle, but they are also transferred so that the expected sporting objectives are guaranteed. This strategic behavior of oligopolistic agents had already been detected in previous works: [2], [4], [18]. However, our study went further – it demonstrated that the greater the differences between the values of a financial or sporting variable observed for two teams, the greater the probability of a player transfer between these two teams in the Spanish championship. We also observed that the greater the differences between these teams (in financial or sporting variables) the greater the values involved in the observed transfers.

These results surpass existing literature as in [5]. Specifically, our results have important implications – the use of transfers as an instrument for the strategic management of clubs cannot be neglected as an important cause for existing imbalances, particularly in terms of competitive balance. Secondly, the observed flow could compromise the financial and sporting viability of smaller teams, which are much more dependent on the surplus resources of larger teams. Thirdly, the structure of the network designed by transfers clearly shows the dependence of the same network on a small group of clubs. This dependence is a serious threat to the integrated objectives of the entire industry, requiring reflection in favour of a more sustainable structure.”

8. Language Editing:

- Prioritize language editing to ensure clarity and readability throughout the manuscript. Clear, concise language is crucial for effective communication.

Response: Thanks for your attention! All your suggestions have been considered.

Reviewer #1: Dear Editor,

I appreciate the opportunity to review the manuscript Exploring Soccer transfers in Spanish League: the hidden role of strategic differences among teams. The aim of this manuscript was “test determinants for the transfer relationships of football players in the Spanish League in the most expensive seasons with records (2017/2018 and 2018/2019 seasons).”. After reading the manuscript, it was possible to observe that:

1 – It is not clear what knowledge gap will be filled.

Thanks for the suggestion. We have clarified the knowledge gap in introduction section:

The main knowledge gap covered by this work is to study the dimensions responsible for the value of transfers from an approach of intense dynamics between clubs using network analysis. This work adds to the high-potential area of social network analysis in sports organizations [1,9,13,17]. In this way, empirical analysis shows relevant findings. First, Spanish soccer transfers are not random, and this evidence reflects the environment of imperfect competition characterizing La Liga. Second, the non-randomness of the transfer process can be associated with an increasing inequality among teams and with a threat to the competitive balance in professional sports. Additionally, we identify that larger point differences, as well as age of the team, balances of transfers and attendances throughout the seasons increase the value of the transfers observed.

2 – There is little theoretical foundation in the introduction.

We have strengthened the theorical foundation in the introduction. We have expanded the explanation of the importance of the study, basing it on existing literature:

“When a player moves from one club to another, his registration is transferred to that club, normally for a fee agreed between the two parties. This is the definition of football transfer, which involves two aspects: the transfer of the player from one team to another and the amount of money agreed between both teams. Transfers in the football world have become a hot topic in academic studies in recent years. The monetary resources involved as well as the growing trend helped in this highlight. The works focused on soccer transfers allow the identification of several lines of development. With no pretensions of exhaustion, we highlight 3 common lines in terms of publications. In a first line, there is the discussion of determinants for such flows, therefore determinants for transfers, either in number of players involved or in the associated amount [1,2,3,4,5,6,7]. In a second line, there are works like Mourao [2,8] that seek to assess the efficiency of transfers considering their own restrictions on the part of the involved determinants. Finally, in a third line, the specificity of the transfers is evaluated in the consideration of focuses, such as the places occupied by some athletes, the nature of the national championships involved or the relationship with certain observed periods [9,11].

In a special way, the study of the determinants of transfers has achieved an important dimension in professional soccer. As Mourao [2] explains, the provisions of Financial fair play have pushed clubs to adopt transfer market strategies to save on relevant expenses and amortize transfers. At the same time, this strategy has increased the level of relevant income in terms of capital gains on player disposals and this effect is stronger for clubs from professional soccer leagues [12]. On the other hand, there are professional soccer leagues that show a notable increase in the monetary resources of player´s transfers. This situation represents an opportunity to analyse the factors that influence transfers. Furthermore, scientific studies have evolved towards the consideration of new statistical tools in the study of determinants in this topic, such as network analysis [13].”

3 – There is no theoretical support for the hypotheses.

Now, the hypotheses are well-defined and theoretically justified:

When we intend to analyze network structures, studies are based on the need to explore not only the dimensions that explain the values observed in flows but also on the need to explore dimensions that explain the existence and meaning of flows in the network.

Thus, we have two groups of hypotheses for the design observed in a network:

- structure hypotheses, where arcs/connection pairs are evaluated, among other more complex structures;

- and the hypotheses of the determinants of values, closer to linear estimation analysis.

Within the second group, we cite the literature that had already been discussed before - [2, 28, 29, 34]. Therefore, for player transfers in the Spanish League, we will analyze sporting performance (measured by the team's number of points in previous seasons) but also the financial performance of that period (evaluated by two variables – the team's budget and the balance of transfers ). But we will also evaluate, for each pair of teams, the respective differential of these variables, as is common in the literature on network analysis. Within the first group, the structure hypotheses, network analysis suggests the assessment of the existence of arcs as well as reciprocal flows [31, 32].

4 – The study sample is old and does not make sense today (more than 4 years have passed).

The maximums observed during the transfer windows listed here, due to the current pandemic, have not yet been exceeded in the most recent transfer windows (p. 8). For this reason, we consider that it is still early to update the data to the most recent sessions, as it could distort the reality of the usual behaviour of the transfer market.

5 – There is no discussion in the article. In the “Empirical analysis” section, a more in-depth discussion with other authors should have taken place, which did not happen. It seems to me that this section is a continuation of the results.

Now, there is a substantial extension of the empirical discussion, also relating the evidence with the previous literature (page 25-26): 

“Let us now discuss these results. As mentioned, with this work we analyzed, in great detail, the seasons of the Spanish championship that had the highest values. We wanted to understand not only the determinants responsible for the structure of the transfer network but also the values involved. We remember that previous works had focused on the importance of studying the network structure for a modern analysis of the phenomenon of professional transfers between sports clubs [18, 19, 20, 21]. Other works had focused on some dimensions of footballer transfers in Spain [1]. However, our work has shown how analyzing the network structure is essential to understanding the phenomenon under analysis, especially the differentiation of values involved.

We emphasize that these results converge with some authors but advance the current state of the art. They converge with [2], [4], [18] in the sense that there is a strategic behavior of clubs in transfers. Players are transferred as an important source of revenue for the club's financial cycle, but they are also transferred so that the expected sporting objectives are guaranteed. This strategic behavior of oligopolistic agents had already been detected in previous works: [2], [4], [18]. However, our study went further – it demonstrated that the greater the differences between the values of a financial or sporting variable observed for two teams, the greater the probability of a player transfer between these two teams in the Spanish championship. We also observed that the greater the differences between these teams (in financial or sporting variables) the greater the values involved in the observed transfers.

These results surpass existing literature as in [5]. Specifically, our results have important implications – the use of transfers as an instrument for the strategic management of clubs cannot be neglected as an important cause for existing imbalances, particularly in terms of competitive balance. Secondly, the observed flow could compromise the financial and sporting viability of smaller teams, which are much more dependent on the surplus resources of larger teams. Thirdly, the structure of the network designed by transfers clearly shows the dependence of the same network on a small group of clubs. This dependence is a serious threat to the integrated objectives of the entire industry, requiring reflection in favour of a more sustainable structure.”

Reviewer #2

Major issues

1. Data

The authors took data for 63 weeks between 1st July 2018 and 1st September 2019. In Europe, there are two transfer windows – one occurs at the beginning of the season (known as the summer window) and another at the mid-season (known as the winter window). The given period covers both the transfer windows of 2018-19 season and the summer window of 2019-20 season. I am unable to understand how the authors claim to analyze the transfers for 2017-18 and 2018-19 seasons using that data. The authors must clarify this.

We would like to clarify that we have made a mistake when writing the analysis period. The true analysis period is 2018-2019 and summer window of 2019-2020 seasons. 

Moreover, according to the authors, the selected seasons are the most expensive two seasons. Doesn’t that make these seasons outliers? Can anything be generalized based on outlier data?.

The chosen seasons are the most expensive but cannot be considered outliers. Rather, this situation responds to an upward tendency in transfers in professional soccer leagues. We have strengthened the theorical foundation in the introduction to explain why the most expensive seasons have been selected. We consider that it is important to understand what happens in these seasons because this can provide relevant findings to explain an upward tendency in transfers that does not only happen in the Spanish league.

2. What did we learn?

It is not clear what the contribution of the paper is. It is well-known that a few clubs dominate all the major football leagues in Europe. The oligopolistic structure exists outside of Spain too – in Germany, France, and England (and to a large extent in Italy). The fact that such an oligopolistic structure distorts the competitive balance is also a well-researched topic.

Thanks for the suggestion. We have clarified the knowledge gap in introduction section:

The main knowledge gap covered by this work is to study the dimensions responsible for the value of transfers from an approach of intense dynamics between clubs using network analysis. This work adds to the high-potential area of social network analysis in sports organizations [1,9,12,16]. In this way, empirical analysis shows relevant findings. First, Spanish soccer transfers are not random, and this evidence reflects the environment of imperfect competition characterizing La Liga. Second, the non-randomness of the transfer process can be associated with an increasing inequality among teams and with a threat to the competitive balance in professional sports. Additionally, we identify that larger point differences, as well as age of the team, balances of transfers and attendances throughout the seasons increase the value of the transfers observed.

We have strengthened the theorical foundation in the introduction. We have expanded the explanation of the importance of the study, basing it on existing literature:

“When a player moves from one club to another, his registration is transferred to that club, normally for a fee agreed between the two parties. This is the definition of football transfer, which involves two aspects: the transfer of the player from one team to another and the amount of money agreed between both teams. Transfers in the football world have become a hot topic in academic studies in recent years. The monetary resources involved as well as the growing trend helped in this highlight. The works focused on soccer transfers allow the identification of several lines of development. With no pretensions of exhaustion, we highlight 3 common lines in terms of publications. In a first line, there is the discussion of determinants for such flows, therefore determinants for transfers, either in number of players involved or in the associated amount [1,2,3,4,5,6,7]. In a second line, there are works like Mourao [2,8] that seek to assess the efficiency of transfers considering their own restrictions on the part of the involved determinants. Finally, in a third line, the specificity of the transfers is evaluated in the consideration of focuses, such as the places occupied by some athletes, the nature of the national championships involved or the relationship with certain observed periods [9,10,11].

In a special way, the study of the determinants of transfers has achieved an important dimension in professional soccer. As Mourao [2] explains, the provisions of Financial fair play have pushed clubs to adopt transfer market strategies to save on relevant expenses and amortize transfers. At the same time, this strategy has increased the level of relevant income in terms of capital gains on player disposals and this effect is stronger for clubs from professional soccer leagues [12]. On the other hand, there are professional soccer leagues that show a notable increase in the monetary resources of player´s transfers. This situation represents an opportunity to analyse the factors that influence transfers. Furthermore, scientific studies have evolved towards the consideration of new statistical tools in the study of determinants in this topic, such as network analysis [13].”

Identification of the fact that teams tend to transfer players with teams that don’t compete for the same spot (in the league table) is also a known fact in football. Typically, in most leagues, rookie players who do well in low-ranked teams are picked up by high-ranked teams. The story of transfer network also involves the B-teams. In Spain, big clubs have their B-teams (and also academies) that play in the second division. Players from these B-teams are often transferred to the mid-ranked teams of the first division. The authors may include such transfers to make the story behind their empirical analysis more cohesive. The main conclusions are now explained in a clearer way.

This work confirmed, through the originality of Network Analysis, the existence of an oligopolistic structure with strategic behaviour in the professional football industry in Spain. Additionally, this work showed that transfers are a vital instrument of this strategy between oligopolists. However, we want to emphasize that this instrument is aimed at temporal consistency, reducing the possibility of mid-table teams being champion teams in Spain. This type of consequence is important for the discussion of the lack of sporting competitiveness in the championship, reduced to a competition whose biggest prize is only within the reach of a small group of teams. 

A different paper would be relevant for a further study only highlighting the B-teams. 

3. Selection of variables

The authors did not justify the selection of variables.

More importantly, they did not consider any sporting variable. Team playing style may be an important factor. For example, the authors may refer to:

McHale, I. G., & Holmes, B. (2023). Estimating transfer fees of professional footballers using advanced performance metrics and machine learning. European Journal of Operational Research, 306(1), 389-399.

The explanatory variables of our model were inspired by the literature presented in the previous section. Still, we recall the underlying reason as well as the supporting literature below.

“The number of points in previous seasons is a variable associated with the value of the team's budget as well as the ability to generate transfers around the club. The use of values from several epochs allows for a more robust associated reading [18, 19]. The average value of the number of points allows you to control possible fluctuations outside the predictable performance of each team.

The variable focused on the transfer balance is a variable that shows the difference between the amount received from transfers made and the amount paid due to transfers that entered the club. In line with the literature consulted [18], higher transfer balances produce a leverage effect on the club's future operations. Once again, the presence of a variable associated with the average value (in this case, the balance of transfers) aims to provide greater robustness to the model.

The average age of the team shows the need to renew the squad in future exercises. As suggested by Samur [22], aging teams increase the probability of incoming transfers in the future.

The total number of spectators reported for each team's home games is a variable that controls the number of points as well as the balance of transfers. Teams with good home game attendances show significant support from the surrounding community. The inclusion of the average attendance at home aims to control possible imbalances that consecutive fluctuations in attendance may suffer [23].”

In relation to incorporating sporting variables, Franceschi, et al. (2023) explain that usage of the player (minutes played or number of appearances), international status and decisiveness (goals and assists) positively and consistently influence the valuation of the football player. On the other hand, Franceschi, et al. (2023) find that the low-score nature of football can explain why the decisiveness of the players estimated through their ability to score and to assist is positively linked with their valuation. Even if the number of specifications in which the performance variables (goals, assists, matches, minutes played, cards) are found to be significant can seem low, these indicators are often tested conjointly even if they are probably correlated, thus they may weaken each other’s influence on the dependent variable. However, we consider it relevant to consider some sporting variables in additional studies.

Several papers have been read and quoted after the suggestion of the reviewers, including the very good references provided by the reviewers. 

4. International transfer

Transfer balance includes the amount received and spent on international transfers, which is a significant part of transfers taking place in each season. Many South American players come to Spain from South American clubs. How can the authors ignore that?

Many players have been transferred to the Spanish championship (as well as other European championships) from the various South American championships. However, the weight of these transfers, both in number and volume, is small compared to the number and total volume of registered transfers. Furthermore, we included in our analysis, despite the previous observation, these transfers from Latin America, which are (like the others) in the “Others” Group (see Figure 1). We also consider that a potential alternative work would involve discriminating the results explained here based on the continents of origin or destination of the observed transfers (please also check our paragraphs regarding further challenges for research).

Minor issue

The paper requires language editing.

Thanks for your support! We have now considered all your suggestions and we are submitting a paper more appropriate to be accepted for publication.

Yours,

The Authors.

---

## [Decision Letter · Decision Letter 1]

12 Dec 2023

PONE-D-23-16261R1Exploring Soccer transfers in Spanish League: the hidden role of strategic differences among teamsPLOS ONE

Dear Dr. Salgado-Barandela,

Thank you for submitting your manuscript to PLOS ONE. After careful consideration, we feel that it has merit but does not fully meet PLOS ONE’s publication criteria as it currently stands. Therefore, we invite you to submit a revised version of the manuscript that addresses the points raised during the review process.

We look forward to receiving your revised manuscript.

Kind regards,

Ricardo Limongi

Academic Editor

PLOS ONE

**Additional Editor Comments:**

Dear Authors,

Thank you for submitting your revised manuscript to our journal. After a thorough evaluation by our reviewers, it has been determined that additional revisions are necessary to enhance the quality and relevance of your work. The topic you are addressing is indeed intriguing and holds potential significance in the field. However, to ensure that your paper makes a substantial contribution to the literature, the following improvements are essential:

1. Network Structure and Transfer Analysis: The reviewers have highlighted the importance of including transfers from the second division, especially from B-teams of championship contender clubs to mid-table clubs. This aspect forms a crucial part of the network structure in your study area and should be incorporated into your analysis to provide a more comprehensive understanding of soccer transfers.

2. Incorporation of Sporting Variables: Considering that your paper focuses on sports, specifically soccer, it is critical to include analysis of sporting variables that influence team playing styles. This addition will enhance the relevance of your study by linking financial aspects of player transfers with the qualitative improvement of teams.

3. Clarification of Knowledge Gap: There is a need for a clearer articulation of the knowledge gap your study aims to fill. This will provide a stronger foundation for your research and its significance within the existing literature.

4. Theoretical Foundation and Hypothesis Support: The introduction requires a more robust theoretical framework. Additionally, the hypotheses presented must be substantiated with adequate theoretical support to strengthen the overall argument of your paper.

5. Updating the Study Sample: The current study sample, based on the 2017/2018 and 2018/2019 seasons, is dated. Updating the sample to include more recent seasons will enhance the relevance and applicability of your findings.

6. Enhanced Discussion Section: The "Empirical Analysis" section of your paper needs a deeper and more critical discussion. This should involve a comparative analysis with other authors' works in the field to contextualize your findings within the broader academic discourse.

We appreciate your efforts in conducting this research and encourage you to undertake these revisions to bring your manuscript to the requisite standard for publication. Your revised manuscript will undergo another round of review. Please feel free to reach out if you need any further guidance or clarification.

We look forward to receiving your revised submission.

Sincerely,

Reviewers' comments:

Reviewer's Responses to Questions

**Comments to the Author**

1. If the authors have adequately addressed your comments raised in a previous round of review and you feel that this manuscript is now acceptable for publication, you may indicate that here to bypass the “Comments to the Author” section, enter your conflict of interest statement in the “Confidential to Editor” section, and submit your "Accept" recommendation.

Reviewer #2: (No Response)

Reviewer #3: All comments have been addressed

2. Is the manuscript technically sound, and do the data support the conclusions?

Reviewer #2: Partly

Reviewer #3: Yes

3. Has the statistical analysis been performed appropriately and rigorously? 

Reviewer #2: No

Reviewer #3: Yes

4. Have the authors made all data underlying the findings in their manuscript fully available?

Reviewer #2: Yes

Reviewer #3: Yes

5. Is the manuscript presented in an intelligible fashion and written in standard English?

Reviewer #2: Yes

Reviewer #3: Yes

6. Review Comments to the Author

Reviewer #2: Report on PONE-D-23-16261R1

Title: Exploring Soccer transfers in Spanish League: the hidden role of strategic differences among teams

The authors have rewritten the introduction and literature review. That did not really improve the paper.

They have not addressed most of my concerns and did not conduct any new analysis based on my suggestions. Consequently, even after the revision, the paper does not make any contribution to the literature.

The topic of the paper is interesting. That is why I want to give the authors another opportunity. They need to do the following to improve the paper:

1. Transfer from the second division, particularly from the B-teams of the championship contender clubs to mid-table clubs, forms an important part of the network structure. Therefore, it is imperative that the authors include these transfers in their analysis.

2. Ultimately, it is a sports paper. Clubs spend money on players to improve the quality of the teams. Therefore, the analysis should include sporting variables that determine the team playing styles.

Reviewer #3: Thank you for the opportunity to review this manuscript. I find it overall interesting, and most of the previous comments have been addressed in the recent manuscript. My further suggestion is to elaborate on your findings and explain how La Liga soccer transfers reflect the hidden role of strategic differences among teams.

7. PLOS authors have the option to publish the peer review history of their article (what does this mean?). If published, this will include your full peer review and any attached files.

Reviewer #2: **Yes: **Sumit Sarkar

Reviewer #3: No

---

## [Author Response · Author response to Decision Letter 1]

16 Feb 2024

Dear Editors of Plos One,

We made a query to PlosOne in relation to the submission of the article: PONE-D-23-16261R1 Exploring Soccer transfers in Spanish League: the hidden role of strategic differences among teams. We have received the response shown below:

“De: plosone <plosone@plos.org>

Enviado: lunes, 8 de enero de 2024 06:43

Para: SALGADO BARANDELA JESYCA MARIA <jesyca.salgado@usc.es>

Asunto: RE: PLOS ONE Decision: Revision required [PONE-D-23-16261R1]

Dear Dr. Salgado-Barandela,

I hope this email finds you well.

Please see below email from Academic Editor in response to your query regarding reviewer comments.

Thank you and if you need further assistance, please let me know. 

Kind regards,

Johnelle Ryan Razo

Straive Editorial Assistant

PLOS ONE | plosone@plos.org

-------------- Original Message ---------------

From: Ricardo Limongi [ricardolimongi@ufg.br]

Sent: 1/6/2024, 3:22 AM

To: plosone@plos.org

Subject: Re: RE: PLOS ONE Decision: Revision required [PONE-D-23-16261R1]

I am in favor of accepting the article. The current version is satisfactory and the authors' argument is coherent, and of course, it would not harm the quality of the article.

The academic editor, Ricardo Limongi, indicates that he is in favour of accepting the article. We would like to know what the next step to take is in relation to its publication. 

At this time, the manuscript appears with the status MajorRevision, in the Editorial Manager and the deadline to submit the review ends on January 24.

Thanks in advance.

Kind regards,

The authors: Paulo Mourao; Jesyca Salgado

---

## [Editor Report · Decision Letter 2]

19 Feb 2024

Exploring Soccer transfers in Spanish League: the hidden role of strategic differences among teams

PONE-D-23-16261R2

Dear Dr. Salgado-Barandela,

We’re pleased to inform you that your manuscript has been judged scientifically suitable for publication and will be formally accepted for publication once it meets all outstanding technical requirements.

Kind regards,

Ricardo Limongi

Academic Editor

PLOS ONE
---

## [Editor Report · Acceptance letter]

27 Mar 2024

PONE-D-23-16261R2 

PLOS ONE

Dear Dr. Salgado-Barandela, 

I'm pleased to inform you that your manuscript has been deemed suitable for publication in PLOS ONE. Congratulations! Your manuscript is now being handed over to our production team.

Kind regards, 

on behalf of

Professor Ricardo Limongi 

Academic Editor

PLOS ONE